# Self-Supervised Aggregation of Diverse Experts for Test-Agnostic Long-Tailed Recognition

**Yifan Zhang**[1]    **Bryan Hooi**[1]    **Lanqing Hong**[2]    **Jiashi Feng**[3]
[1]National University of Singapore    [2]Huawei Noah's Ark Lab    [3]ByteDance
`yifan.zhang@u.nus.edu, jshfeng@gmail.com`

## Abstract

Existing long-tailed recognition methods, aiming to train class-balanced models from long-tailed data, generally assume the models would be evaluated on the uniform test class distribution. However, practical test class distributions often violate this assumption (*e.g.,* being either long-tailed or even inversely long-tailed), which may lead existing methods to fail in real applications. In this paper, we study a more practical yet challenging task, called *test-agnostic long-tailed recognition*, where the training class distribution is long-tailed while the test class distribution is *agnostic and not necessarily uniform*. In addition to the issue of class imbalance, this task poses another challenge: the class distribution shift between the training and test data is unknown. To tackle this task, we propose a novel approach, called *Self-supervised Aggregation of Diverse Experts*, which consists of two strategies: (i) a new skill-diverse expert learning strategy that trains multiple experts from a single and stationary long-tailed dataset to separately handle different class distributions; (ii) a novel test-time expert aggregation strategy that leverages self-supervision to aggregate the learned multiple experts for handling unknown test class distributions. We theoretically show that our self-supervised strategy has a provable ability to simulate test-agnostic class distributions. Promising empirical results demonstrate the effectiveness of our method on both vanilla and test-agnostic long-tailed recognition. The source code is available at https://github.com/Vanint/SADE-AgnosticLT.

## 1   Introduction

Real-world visual recognition datasets typically exhibit a long-tailed distribution, where a few classes contain numerous samples (called head classes), but the others are associated with only a few instances (called tail classes) [24, 33]. Due to the class imbalance, the trained model is easily biased towards head classes and perform poorly on tail classes [2, 58]. To tackle this issue, numerous studies have explored long-tailed recognition for learning well-performing models from imbalanced data [20, 56].

Most existing long-tailed studies [3, 9, 10, 48, 52] assume the test class distribution is uniform, *i.e.,* each class has an equal amount of test data. Therefore, they develop various techniques, *e.g.,* class re-sampling [13, 18, 25, 55], cost-sensitive learning [11, 36, 41, 47] or ensemble learning [2, 13, 27, 53], to re-balance the model performance on different classes for fitting the uniform class distribution. However, this assumption does not always hold in real applications, where actual test data may follow any kind of class distribution, being either uniform, long-tailed, or even inversely long-tailed to the training data (cf. Figure 1(a)). For example, one may train a recognition model for autonomous cars based on the training data collected from city areas, where pedestrians are majority classes and stone obstacles are minority classes. However, when the model is deployed to mountain areas, the pedestrians become the minority while the stones become the majority. In this case, the test class distribution is inverse to the training one, and existing methods may perform poorly.

36th Conference on Neural Information Processing Systems (NeurIPS 2022).

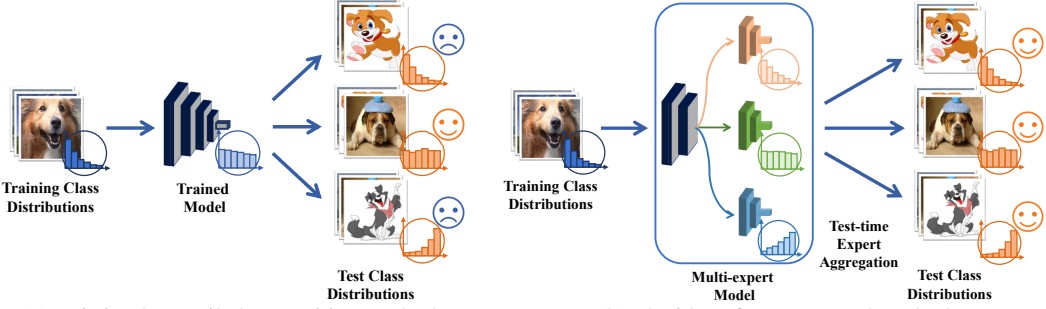

(a) Existing long-tailed recognition methods      (b) The idea of our proposed method

Figure 1: Illustration of test-agnostic long-tailed recognition. (a) Existing long-tailed recognition methods aim to train models that perform well on test data with the uniform class distribution. However, the resulting models may fail to handle practical test class distributions that skew arbitrarily. (b) Our method seeks to learn a multi-expert model from a single long-tailed training set, where different experts are skilled in handling different class distributions, respectively. By reasonably aggregating these experts at test time, our method is able to handle unknown test class distributions.

To address the issue of varying class distributions, as the first research attempt, LADE [17] assumes the test class distribution to be known and uses the knowledge to post-adjust model predictions. However, the actual test class distribution is usually unknown a priori, making LADE not applicable in practice. Therefore, we study a more realistic yet challenging problem, namely *test-agnostic long-tailed recognition*, where the training class distribution is long-tailed while the test distribution is *agnostic*. To tackle this problem, motivated by the idea of "divide and conquer", we propose to learn multiple experts with *diverse* skills that excel at handling different class distributions (cf. Figure 1(b)). As long as these skill-diverse experts can be aggregated suitably at test time, the multi-expert model would manage to handle the unknown test class distribution. Following this idea, we develop a novel approach, namely *Self-supervised Aggregation of Diverse Experts* (SADE).

The first challenge for SADE is how to learn multiple *diverse* experts from a *single* and *stationary* long-tailed training dataset. To handle this challenge, we empirically evaluate existing long-tailed methods in this task, and find that the models trained by existing methods have a *simulation correlation between the learned class distribution and the training loss function*. That is, the models learned by various losses are skilled in handling class distributions with different skewness. For example, the model trained with the conventional softmax loss simulates the long-tailed training class distribution, while the models obtained from existing long-tailed methods are good at the uniform class distribution. Inspired by this finding, SADE presents a simple but effective skill-diverse expert learning strategy to generate experts with different distribution preferences from a single long-tailed training distribution. Here, various experts are trained with different expertise-guided objective functions to deal with different class distributions, respectively. As a result, the learned experts are more diverse than previous multi-expert long-tailed methods [49, 63], leading to better ensemble performance, and in aggregate simulate a wide spectrum of possible class distributions.

The other challenge is how to aggregate these skill-diverse experts for handling test-agnostic class distributions based on only *unlabeled test data*. To tackle this challenge, we empirically investigate the property of different experts, and observe that there is *a positive correlation between expertise and prediction stability*, *i.e.,* stronger experts have higher prediction consistency between different perturbed views of samples from their favorable classes. Motivated by this finding, we develop a novel self-supervised strategy, namely prediction stability maximization, to adaptively aggregate experts based on only unlabeled test data. We theoretically show that maximizing the prediction stability enables SADE to learn an aggregation weight that maximizes the mutual information between the predicted label distribution and the true class distribution. In this way, the resulting model is able to simulate unknown test class distributions.

We empirically verify the superiority of SADE on both vanilla and test-agnostic long-tailed recognition. Specifically, SADE achieves promising performance on vanilla long-tailed recognition under all benchmark datasets. For instance, SADE achieves 58.8% accuracy on ImageNet-LT with more than 2% accuracy gain over previous state-of-the-art ensemble long-tailed methods, *i.e.,* RIDE [49] and ACE [2]. More importantly, SADE is the first long-tailed approach that is able to handle various test-agnostic class distributions without knowing the true class distribution of test data in advance. Note that SADE even outperforms LADE [17] that uses knowledge of the test class distribution.

Compared to previous long-tailed methods (*e.g.,* LADE [17] and RIDE [49]), our method offers the following advantages: (i) SADE does not assume the test class distribution to be known, and provides the first practical approach to handling test-agnostic long-tailed recognition; (ii) SADE develops a simple diversity-promoting strategy to learn skill-diverse experts from a single and stationary long-tailed dataset; (iii) SADE presents a novel self-supervised strategy to aggregate skill-diverse experts at test time, by maximizing prediction consistency between unlabeled test samples' perturbed views; (iv) the presented self-supervised strategy has a provable ability to simulate test-agnostic class distributions, which opens the opportunity for tackling unknown class distribution shifts at test time.

## 2  Related Work

**Long-tailed recognition**    Existing long-tailed recognition methods, related to our study, can be categorized into three types: class re-balancing, logit adjustment and ensemble learning. Specifically, class re-balancing resorts to re-sampling [4, 13, 18, 25] or cost-sensitive learning [3, 10, 16, 61] to balance different classes during model training. Logit adjustment [17, 33, 37, 43] adjusts models' output logits via the label frequencies of training data at inference time, for obtaining a large relative margin between head and tail classes. Ensemble-based methods [2, 13, 53, 63], *e.g.,* RIDE [49], are based on multiple experts, which seek to capture heterogeneous knowledge, followed by ensemble aggregation. More discussions on the difference between our method and RIDE [49] are provided in Appendix D.3. Regarding test-agnostic long-tailed recognition, LADE [17] assumes the test class distribution is available and uses it to post-adjust model predictions. However, the true test class distribution is usually unknown a priori, making LADE inapplicable. In contrast, our method does not rely on the true test distribution for handling this problem, but presents a novel self-supervised strategy to aggregate skill-diverse experts at test time for test-agnostic class distributions. Moreover, some ensemble-based long-tailed methods [39] aggregate experts based on a *labeled* uniform validation set. However, as the test class distribution could be different from the validation one, simply aggregating experts on the validation set is unable to handle test-agnostic long-tailed recognition.

**Test-time training**    Test-time training [23, 26, 30, 40, 46] is a transductive learning paradigm for handling distribution shifts [28, 32, 34, 38, 45, 59] between training and test data, and has been applied with success to out-of-domain generalization [19, 35] and dynamic scene deblurring [6]. In this study, we explore this paradigm to handle test-agnostic long-tailed recognition, where the issue of class distribution shifts is the main challenge. However, most existing test-time training methods seek to handle covariate distribution shifts instead of class distribution shifts, so simply leveraging them cannot resolve test-agnostic long-tailed recognition, as shown in our experiment (cf. Table 9).

## 3  Problem Formulation

Long-tailed recognition aims to learn a well-performing classification model from a training dataset with long-tailed class distribution. Let $\mathcal{D}_s = \{x_i, y_i\}_{i=1}^{n_s}$ denote the long-tailed training set, where $y_i$ is the class label of the sample $x_i$. The total number of training data over $C$ classes is $n_s = \sum_{k=1}^{C} n_k$, where $n_k$ denotes the number of samples in class $k$. Without loss of generality, we follow a common assumption [17, 25] that the classes are sorted by cardinality in decreasing order (*i.e.,* if $i_1 < i_2$, then $n_{i_1} \geq n_{i_2}$), and $n_1 \gg n_C$. The imbalance ratio is defined as $\max(n_k)/\min(n_k) = n_1/n_C$. The test data $\mathcal{D}_t = \{x_j, y_j\}_{j=1}^{n_t}$ is defined in a similar way.

Most existing long-tailed recognition methods assume the test class distribution is uniform (*i.e.,* $p_t(y) = 1/C$), and seek to train models from the long-tailed training distribution $p_s(y)$ to perform well on the uniform test distribution. However, such an assumption does not always hold in practice. The actual test class distribution in real-world applications may also be long-tailed (*i.e.,* $p_t(y) = p_s(y)$), or even inversely long-tailed to the training data (*i.e.,* $p_t(y) = \text{inv}(p_s(y))$). Here, $\text{inv}(\cdot)$ indicates that the order of the long tail on classes is flipped. As a result, the models learned by existing methods may fail when the actual test class distribution is different from the assumed one. To address this, we propose to study a more practical yet challenging long-tailed problem, *i.e.,* **Test-agnostic Long-tailed Recognition**. This task aims to learn a recognition model from long-tailed training data, where the resulting model would be evaluated on multiple test sets that follow different class distributions. This task is challenging due to the integration of two challenges: (1) the severe class imbalance in the training data makes it difficult to train models; (2) unknown class distribution shifts between training and test data (*i.e.,* $p_t(y) \neq p_s(y)$) makes models hard to generalize.

Table 1: Accuracy of existing long-tailed (LT) methods on ImageNet-LT with various test class distributions, including uniform, forward and backward LT distributions with imbalance ratios of 10 and 50, respectively. The results show that each method strives to simulate a specific class distribution in terms of many-shot, medium-shot and few-shot classes, which does not change when the test class distribution varies. The corresponding visualization results are reported in Figure 5 in Appendix D.4.

| Test class distribution | Softmax | | | Balanced Softmax [21] | | | LADE w/o prior [17] | | |
|---|---|---|---|---|---|---|---|---|---|
| | Many | Medium | Few | Many | Medium | Few | Many | Medium | Few |
| Forward-LT-50 | 67.5 | 41.7 | 14.0 | 63.5 | 47.8 | 37.5 | 63.5 | 46.4 | 33.1 |
| Forward-LT-10 | 68.2 | 40.9 | 14.0 | 64.1 | 48.2 | 31.2 | 64.7 | 47.1 | 32.2 |
| Uniform | 68.1 | 41.5 | 14.0 | 64.1 | 48.2 | 33.4 | 64.4 | 47.7 | 34.3 |
| Backward-LT-10 | 67.4 | 41.9 | 13.9 | 63.4 | 49.1 | 33.6 | 64.4 | 48.2 | 34.2 |
| Backward-LT-50 | 70.9 | 41.1 | 13.8 | 66.5 | 48.4 | 33.2 | 66.3 | 47.8 | 34.0 |

## 4 Method

To tackle the above problem, inspired by the idea of "divide and conquer", we propose to learn multiple skill-diverse experts that excel at handling different class distributions. By reasonably fusing these experts at test time, the multi-expert model would manage to handle unknown class distribution shifts and resolve test-agnostic long-tailed recognition. Following this idea, we develop a novel Self-supervised Aggregation of Diverse Experts (SADE) approach. Specifically, SADE consists of two innovative strategies: (1) *learning skill-diverse experts* from a single long-tailed training dataset; (2) *test-time aggregating experts with self-supervision* to handle test-agnostic class distributions.

### 4.1 Skill-diverse Expert Learning

As shown in Figure 2, SADE builds a three-expert model that comprises two components: (1) an expert-shared backbone $f_\theta$; (2) independent expert networks $E_1$, $E_2$ and $E_3$. When training the model, the key challenge is how to learn skill-diverse experts from a single and stationary long-tailed training dataset. Existing ensemble-based long-tailed methods [13, 49] seek to train experts for the uniform test class distribution, and hence the trained experts are not differentiated sufficiently for handling various class distributions (refer to Table 6 for an example). To tackle this challenge, we first empirically investigate existing long-tailed methods in this task. From Table 1, we find that there is a *simulation correlation between the learned class distribution and the training loss function*. That is, the models learned by different losses are good at dealing with class distributions with different skewness. For instance, the model trained with the softmax loss is good at the long-tailed distribution, while the models obtained from long-tailed methods are skilled in the uniform distribution.

Motivated by this finding, we develop a simple skill-diverse expert learning strategy to generate experts with different distribution preferences. To be specific, the forward expert $E_1$ seeks to be good at the long-tailed class distribution and performs well on many-shot classes. The uniform expert $E_2$ strives to be skilled in the uniform distribution. The backward expert $E_3$ aims at the inversely long-tailed distribution and performs well on few-shot classes. Here, the forward and backward experts are necessary since they span a wide spectrum of possible class distributions, while the uniform expert ensures retaining high accuracy on the uniform distribution. To this end, we use three different expertise-guided losses to train the three experts, respectively.

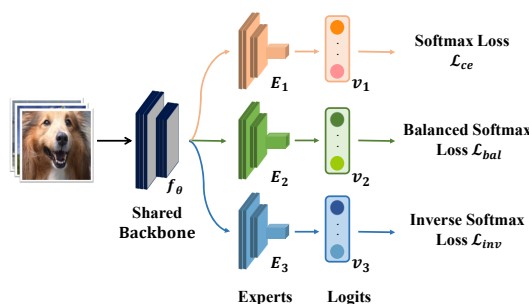

Figure 2: The scheme of SADE with three experts, where different experts are trained with different expertise-guided losses.

**The forward expert $E_1$** We use the softmax cross-entropy loss to train this expert, so that it directly simulates the original long-tailed training class distribution:

$$\mathcal{L}_{ce} = \frac{1}{n_s} \sum_{x_i \in \mathcal{D}_s} -y_i \log \sigma(v_1(x_i)), \tag{1}$$

where $v_1(\cdot)$ is the output logits of the forward expert $E_1$, and $\sigma(\cdot)$ is the softmax function.

**The uniform expert** $E_2$    We aim to train this expert to simulate the uniform class distribution. Inspired by the effectiveness of logit adjusted losses for long-tailed recognition [33], we resort to the balanced softmax loss [21]. Specifically, let $\hat{y}^k = \frac{\exp(v^k)}{\sum_{c=1}^{C} \exp(v^c)}$ be the prediction probability. The balanced softmax adjusts the prediction probability by compensating for the long-tailed class distribution with the prior of training label frequencies: $\hat{y}^k = \frac{\pi^k \exp(v^k)}{\sum_{c=1}^{C} \pi^c \exp(v^c)} = \frac{\exp(v^k + \log \pi^k)}{\sum_{c=1}^{C} \exp(v^c + \log \pi^c)}$, where $\pi^k = \frac{n_k}{n}$ denotes the training label frequency of class $k$. Then, given $v_2(\cdot)$ as the output logits of the expert $E_2$, the balanced softmax loss for the expert $E_2$ is defined as:

$$\mathcal{L}_{bal} = \frac{1}{n_s} \sum_{x_i \in \mathcal{D}_s} -y_i \log \sigma(v_2(x_i) + \log \pi). \tag{2}$$

Intuitively, by adjusting logits to compensate for the long-tailed distribution with the prior $\pi$, this loss enables $E_2$ to output class-balanced predictions that simulate the uniform distribution.

**The backward expert** $E_3$    We seek to train this expert to simulate the inversely long-tailed class distribution. To this end, we propose a new *inverse softmax loss*, based on the same rationale of logit adjusted losses [21, 33]. Specifically, we adjust the prediction probability by: $\hat{y}^k = \frac{\exp(v^k + \log \pi^k - \log \bar{\pi}^k)}{\sum_{c=1}^{C} \exp(v^c + \log \pi^c - \log \bar{\pi}^c)}$, where the inverse training prior $\bar{\pi}$ is obtained by inverting the order of training label frequencies $\pi$. Then, the new inverse softmax loss for the expert $E_3$ is defined as:

$$\mathcal{L}_{inv} = \frac{1}{n_s} \sum_{x_i \in \mathcal{D}_s} -y_i \log \sigma(v_3(x_i) + \log \pi - \lambda \log \bar{\pi}), \tag{3}$$

where $v_3(\cdot)$ denotes the output logits of $E_3$ and $\lambda$ is a hyper-parameter. Intuitively, this loss adjusts logits to compensate for the long-tailed distribution with $\pi$, and further applies reverse adjustment with $\bar{\pi}$. This enables $E_3$ to simulate the inversely long-tailed distribution (cf. Table 6 for verification).

## 4.2   Test-time Self-supervised Aggregation

Based on the skill-diverse learning strategy, the three experts in SADE are skilled in different class distributions. The remaining challenge is how to fuse them to deal with unknown test class distributions. A basic principle for expert aggregation is that the experts should play a bigger role in situations where they have expertise. Nevertheless, how to detect strong experts for unknown test class distribution remains unknown. Our key insight is that strong experts should be more stable in predicting the samples from their skilled classes, even though these samples are perturbed.

**Empirical observation**    To verify this hypothesis, we estimate the prediction stability of experts by comparing the cosine similarity between their predictions for a sample's two augmented views. Here, the data views are generated by the data augmentation techniques in MoCo v2 [5]. From Table 2, we find that there is a *positive correlation between expertise and prediction stability*, *i.e.,* stronger experts have higher prediction similarity between different views of samples from their favorable classes. Following this finding, we propose to explore the relative prediction stability to detect strong experts and weight experts for the unknown test class distribution. Consequently, we develop a novel self-supervised strategy, namely prediction stability maximization.

**Prediction stability maximization**    This strategy learns aggregation weights for experts (with frozen parameters) by maximizing model prediction stability for unlabeled test samples. As shown in Figure 3, the method comprises three major components as follows.

*Data view generation*    For a given sample $x$, we conduct two stochastic data augmentations to generate the sample's two views, *i.e.,* $x^1$ and $x^2$. Here, we use the same augmentation techniques as the advanced contrastive learning method, *i.e.,* MoCo v2 [5], which has been shown effective in self-supervised learning.

*Learnable aggregation weight*    Given the output logits of three experts $(v_1, v_2, v_3) \in \mathbb{R}^{3 \times C}$, we aggregate experts with a learnable aggregation weight $w = [w_1, w_2, w_3] \in \mathbb{R}^3$ and obtain the final softmax prediction by $\hat{y} = \sigma(w_1 \cdot v_1 + w_2 \cdot v_2 + w_3 \cdot v_3)$, where $w$ is normalized before aggregation, *i.e.,* $w_1 + w_2 + w_3 = 1$.

Table 2: Prediction stability of experts in terms of the cosine similarity between their predictions of a sample's two views. Note that expert $E_1$ is good at many-shot classes and expert $E_3$ is skilled in few-shot classes. The experts tend to have better prediction consistency for the samples from their skilled classes. Here, the imbalance ratio of CIFAR100-LT is 100.

| Model | Cosine similarity between view predictions | | | | | |
| | ImageNet-LT | | | CIFAR100-LT | | |
| | Many | Med. | Few | Many | Med. | Few |
|---|---|---|---|---|---|---|
| Expert $E_1$ | 0.60 | 0.48 | 0.43 | 0.28 | 0.22 | 0.20 |
| Expert $E_2$ | 0.56 | 0.50 | 0.45 | 0.25 | 0.21 | 0.19 |
| Expert $E_3$ | 0.52 | 0.53 | 0.58 | 0.22 | 0.23 | 0.25 |

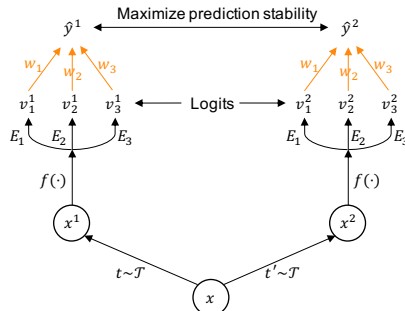

Figure 3: The scheme of test-time self-supervised aggregation. Two data augmentations sampled from the same family of augmentations ($t \sim \mathcal{T}$ and $t' \sim \mathcal{T}$) are applied to obtain two data views.

*Objective function*   Given the view predictions of unlabeled test data, we maximize the prediction stability based on the cosine similarity between the view predictions:

$$\max_w \mathcal{S}, \ \ \text{where} \ \ \mathcal{S} = \frac{1}{n_t} \sum_{x \in \mathcal{D}_t} \hat{y}^1 \cdot \hat{y}^2. \tag{4}$$

Here, $\hat{y}^1$ and $\hat{y}^2$ are normalized by the softmax function. In test-time training, only the aggregation weight $w$ is updated. Since stronger experts have higher prediction similarity for their skilled classes, maximizing the prediction stability $\mathcal{S}$ would learn higher weights for stronger experts regarding the unknown test class distribution. Moreover, the self-supervised aggregation strategy can be conducted in an online manner for streaming test data. The pseudo-code of SADE is provided in Appendix B.

**Theoretical Analysis**   We then theoretically analyze the prediction stability maximization strategy to conceptually understand why it works. To this end, we first define the random variables of predictions and labels as $\hat{Y} \sim p(\hat{y})$ and $Y \sim p_t(y)$. We have the following result:

**Theorem 1.** *The prediction stability $\mathcal{S}$ is positive proportional to the mutual information between the predicted label distribution and the test class distribution $I(\hat{Y}; Y)$, and negative proportional to the prediction entropy $H(\hat{Y})$:*

$$\mathcal{S} \ \propto \ I(\hat{Y}; Y) - H(\hat{Y}).$$

Please refer to Appendix A for proofs. According to Theorem 1, maximizing the prediction stability $\mathcal{S}$ enables SADE to learn an aggregation weight that maximizes the mutual information between the predicted label distribution $p(\hat{y})$ and the test class distribution $p_t(y)$, as well as minimizing the prediction entropy. Since minimizing entropy helps to improve the confidence of the classifier output [12], the aggregation weight is learned to simulate the test class distribution $p_t(y)$ and increase the prediction confidence. This property intuitively explains why our method has the potential to tackle the challenging task of test-agnostic long-tailed recognition at test time.

## 5   Experiments

In this section, we first evaluate the superiority of SADE on both vanilla and test-agnostic long-tailed recognition. We then verify the effectiveness of SADE in terms of its two strategies, *i.e.,* skill-diverse expert learning and test-time self-supervised aggregation. More ablation studies are reported in appendices. Here, we begin with the experimental settings.

### 5.1   Experimental Setups

**Datasets**   We use four benchmark datasets (*i.e.,* ImageNet-LT [31], CIFAR100-LT [3], Places-LT [31], and iNaturalist 2018 [44]) to simulate real-world long-tailed class distributions. Their data statistics and imbalance ratios are summarized in Appendix C.1. The imbalance ratio is defined as $\max n_j / \min n_j$, where $n_j$ denotes the data number of class $j$. Note that CIFAR100-LT has three variants with different imbalance ratios.

Table 3: Top-1 accuracy on CIFAR100-LT, Places-LT and iNaturalist 2018, where the test class distribution is uniform. More results on three class sub-groups are reported in Appendix D.1.

| (a) CIFAR100-LT | | | |
| --- | --- | --- | --- |
| Imbalance Ratio | 10 | 50 | 100 |
| Softmax | 59.1 | 45.6 | 41.4 |
| BBN [63] | 59.8 | 49.3 | 44.7 |
| Causal [42] | 59.4 | 48.8 | 45.0 |
| Balanced Softmax [21] | 61.0 | 50.9 | 46.1 |
| MiSLAS [62] | 62.5 | 51.5 | 46.8 |
| LADE [17] | 61.6 | 50.1 | 45.6 |
| RIDE [49] | 61.8 | 51.7 | 48.0 |
| SADE (ours) | **63.6** | **53.9** | **49.8** |

| (b) Places-LT | |
| --- | --- |
| Method | Top-1 accuracy |
| Softmax | 31.4 |
| Causal [42] | 32.2 |
| Balanced Softmax [21] | 39.4 |
| MiSLAS [62] | 38.3 |
| LADE [17] | 39.2 |
| RIDE [49] | 40.3 |
| SADE (ours) | **40.9** |

| (c) iNaturalist 2018 | |
| --- | --- |
| Method | Top-1 accuracy |
| Softmax | 64.7 |
| Causal [42] | 64.4 |
| Balanced Softmax [21] | 70.6 |
| MiSLAS [62] | 70.7 |
| LADE [17] | 69.3 |
| RIDE [49] | 71.8 |
| SADE (ours) | **72.9** |

**Baselines** We compare SADE with state-of-the-art long-tailed methods, including two-stage methods (Decouple [25], MiSLAS [62]), logit-adjusted training (Balanced Softmax [21], LADE [17]), ensemble learning (BBN [63], ACE [2], RIDE [49]), classifier design (Causal [42]), and representation learning (PaCo [8]). Note that LADE uses the prior of test class distribution for post-adjustment (although it is unavailable in practice), while all other methods do not use this prior.

**Evaluation protocols** In test-agnostic long-tailed recognition, following LADE [17], the models are evaluated on multiple sets of test data that follow different class distributions, in terms of micro accuracy. Same as LADE [17], we construct three kinds of test class distributions, *i.e.,* the uniform distribution, forward long-tailed distributions as training data, and backward long-tailed distributions. In the backward ones, the order of the long tail on classes is flipped. More details of test data construction are provided in Appendix C.2. Besides, we also evaluate methods on vanilla long-tailed recognition [25, 31], where the models are evaluated on the uniform test class distribution. Here, the accuracy on three class sub-groups is also reported, *i.e.,* many-shot classes (more than 100 training images), medium-shot classes (20∼100 images) and few-shot classes (less than 20 images).

**Implementation details** We use the same setup for all the baselines and our method. Specifically, following [17, 49], we use ResNeXt-50 for ImageNet-LT, ResNet-32 for CIFAR100-LT, ResNet-152 for Places-LT and ResNet-50 for iNaturalist 2018 as backbones, respectively. Moreover, we adopt the cosine classifier for prediction on all datasets. If not specified, we use the SGD optimizer with the momentum of 0.9 for training 200 epochs and set the initial learning rate as 0.1 with linear decay. We set $\lambda=2$ for ImageNet-LT and CIFAR100-LT, and $\lambda=1$ for the remaining datasets. During test-time training, we train the aggregation weights for 5 epochs with the batch size 128, where we use the same optimizer and learning rate as the training phase. More implementation details and the hyper-parameter statistics are reported in Appendix C.3.

## 5.2 Superiority on Vanilla Long-tailed Recognition

This subsection compares SADE with state-of-the-art long-tailed methods on vanilla long-tailed recognition. Specifically, as shown in Tables 3-4, Softmax trains the model with only cross-entropy, so it simulates the long-tailed training distribution and performs well on many-shot classes. However, it performs poorly on medium-shot and few-shot classes, leading to worse overall performance. In contrast, existing long-tailed methods (*e.g.,* Decouple, Causal) seek to simulate the uniform class distribution, so their performance is more class-balanced, leading to better overall performance. However, as these methods mainly seek balanced performance, they in-

Table 4: Top-1 accuracy on ImageNet-LT.

| Method | Many | Med. | Few | All |
| --- | --- | --- | --- | --- |
| Softmax | 68.1 | 41.5 | 14.0 | 48.0 |
| Decouple-LWS [25] | 61.8 | 47.6 | 30.9 | 50.8 |
| Causal [42] | 64.1 | 45.8 | 27.2 | 50.3 |
| Balanced Softmax [21] | 64.1 | 48.2 | 33.4 | 52.3 |
| MiSLAS [62] | 62.0 | 49.1 | 32.8 | 51.4 |
| LADE [17] | 64.4 | 47.7 | 34.3 | 52.3 |
| PaCo [8] | 63.2 | 51.6 | 39.2 | 54.4 |
| ACE [2] | **71.7** | 54.6 | 23.5 | 56.6 |
| RIDE [49] | 68.0 | 52.9 | 35.1 | 56.3 |
| SADE (ours) | 66.5 | **57.0** | **43.5** | **58.8** |

evitably sacrifice the performance on many-shot classes. To address this, RIDE and ACE explore ensemble learning for long-tailed recognition and achieve better performance on tail classes without sacrificing the head-class performance. In comparison, based on the increasing expert diversity derived from skill-diverse expert learning, our method performs the best on all datasets, *e.g.,* with more than 2% accuracy gain on ImageNet-LT compared to RIDE and ACE. These results demonstrate the superiority of SADE over the compared methods that are particularly designed for the uniform test class distribution. Note that SADE also outperforms baselines in experiments with stronger data augmentation (*i.e.,* RandAugment [7]) and other architectures, as reported in Appendix D.1.

Table 5: Top-1 accuracy on long-tailed datasets with various unknown test class distributions. "Prior" indicates that the test class distribution is used as prior knowledge. "Uni." denotes the uniform distribution. "IR" indicates the imbalance ratio. "BS" denotes the balanced softmax [21].

(a) ImageNet-LT

| Method | Prior | Forward-LT | | | | | Uni. | Backward-LT | | | | |
|---|---|---|---|---|---|---|---|---|---|---|---|---|
| | | 50 | 25 | 10 | 5 | 2 | 1 | 2 | 5 | 10 | 25 | 50 |
| Softmax | ✗ | 66.1 | 63.8 | 60.3 | 56.6 | 52.0 | 48.0 | 43.9 | 38.6 | 34.9 | 30.9 | 27.6 |
| BS | ✗ | 63.2 | 61.9 | 59.5 | 57.2 | 54.4 | 52.3 | 50.0 | 47.0 | 45.0 | 42.3 | 40.8 |
| MiSLAS | ✗ | 61.6 | 60.4 | 58.0 | 56.3 | 53.7 | 51.4 | 49.2 | 46.1 | 44.0 | 41.5 | 39.5 |
| LADE | ✗ | 63.4 | 62.1 | 59.9 | 57.4 | 54.6 | 52.3 | 49.9 | 46.8 | 44.9 | 42.7 | 40.7 |
| LADE | ✓ | 65.8 | 63.8 | 60.6 | 57.5 | 54.5 | 52.3 | 50.4 | 48.8 | 48.6 | 49.0 | 49.2 |
| RIDE | ✗ | 67.6 | 66.3 | 64.0 | 61.7 | 58.9 | 56.3 | 54.0 | 51.0 | 48.7 | 46.2 | 44.0 |
| SADE | ✗ | **69.4** | **67.4** | **65.4** | **63.0** | **60.6** | **58.8** | **57.1** | **55.5** | **54.5** | **53.7** | **53.1** |

(b) CIFAR100-LT (IR100)

| Method | Prior | Forward-LT | | | | | Uni. | Backward-LT | | | | |
|---|---|---|---|---|---|---|---|---|---|---|---|---|
| | | 50 | 25 | 10 | 5 | 2 | 1 | 2 | 5 | 10 | 25 | 50 |
| Softmax | ✗ | 63.3 | 62.0 | 56.2 | 52.5 | 46.4 | 41.4 | 36.5 | 30.5 | 25.8 | 21.7 | 17.5 |
| BS | ✗ | 57.8 | 55.5 | 54.2 | 52.0 | 48.7 | 46.1 | 43.6 | 40.8 | 38.4 | 36.3 | 33.7 |
| MiSLAS | ✗ | 58.8 | 57.2 | 55.2 | 53.0 | 49.6 | 46.8 | 43.6 | 40.1 | 37.7 | 33.9 | 32.1 |
| LADE | ✗ | 56.0 | 55.5 | 52.8 | 51.0 | 48.0 | 45.6 | 43.2 | 40.0 | 38.3 | 35.5 | 34.0 |
| LADE | ✓ | 62.6 | 60.2 | 55.6 | 52.7 | 48.2 | 45.6 | 43.8 | 41.1 | 41.5 | 40.7 | 41.6 |
| RIDE | ✗ | 63.0 | 59.9 | 57.0 | 53.6 | 49.4 | 48.0 | 42.5 | 38.1 | 35.4 | 31.6 | 29.2 |
| SADE | ✗ | **65.9** | **62.5** | **58.3** | **54.8** | **51.1** | **49.8** | **46.2** | **44.7** | **43.9** | **42.5** | **42.4** |

(c) Places-LT

| Method | Prior | Forward-LT | | | | | Uni. | Backward-LT | | | | |
|---|---|---|---|---|---|---|---|---|---|---|---|---|
| | | 50 | 25 | 10 | 5 | 2 | 1 | 2 | 5 | 10 | 25 | 50 |
| Softmax | ✗ | 45.6 | 42.7 | 40.2 | 38.0 | 34.1 | 31.4 | 28.4 | 25.4 | 23.4 | 20.8 | 19.4 |
| BS | ✗ | 42.7 | 41.7 | 41.3 | 41.0 | 40.0 | 39.4 | 38.5 | 37.8 | 37.1 | 36.2 | 35.6 |
| MiSLAS | ✗ | 40.9 | 39.7 | 39.5 | 39.6 | 38.8 | 38.3 | 37.3 | 36.7 | 35.8 | 34.7 | 34.4 |
| LADE | ✗ | 42.8 | 41.5 | 41.2 | 40.8 | 39.8 | 39.2 | 38.1 | 37.6 | 36.9 | 36.0 | 35.7 |
| LADE | ✓ | 46.3 | 44.2 | 42.2 | 41.2 | 39.7 | 39.4 | 39.2 | 39.9 | 40.9 | **42.4** | **43.0** |
| RIDE | ✗ | 43.1 | 41.8 | 41.6 | 42.0 | 41.0 | 40.3 | 39.6 | 38.7 | 38.2 | 37.0 | 36.9 |
| SADE | ✗ | **46.4** | **44.9** | **43.3** | **42.6** | **41.3** | **40.9** | **40.6** | **41.1** | **41.4** | 42.0 | 41.6 |

(d) iNaturalist 2018

| Method | Prior | Forward-LT | | Uni. | Backward-LT | |
|---|---|---|---|---|---|---|
| | | 3 | 2 | 1 | 2 | 3 |
| Softmax | ✗ | 65.4 | 65.5 | 64.7 | 64.0 | 63.4 |
| BS | ✗ | 70.3 | 70.5 | 70.6 | 70.6 | 70.8 |
| MiSLAS | ✗ | 70.8 | 70.8 | 70.7 | 70.7 | 70.2 |
| LADE | ✗ | 68.4 | 69.0 | 69.3 | 69.6 | 69.5 |
| LADE | ✓ | ✗ | 69.1 | 69.3 | 70.2 | ✗ |
| RIDE | ✗ | 71.5 | 71.9 | 71.8 | 71.9 | 71.8 |
| SADE | ✗ | **72.3** | **72.5** | **72.9** | **73.5** | **73.3** |

Table 6: Performance of each expert on the uniform test distribution, where the imbalance ratio of CIFAR100-LT is 100. The results show that our proposed method learns multiple experts with higher skill diversity, which leads to better ensemble performance.

| Model | RIDE [49] | | | | | | | | SADE (ours) | | | | | | | |
|---|---|---|---|---|---|---|---|---|---|---|---|---|---|---|---|---|
| | ImageNet-LT | | | | CIFAR100-LT | | | | ImageNet-LT | | | | CIFAR100-LT | | | |
| | Many | Med. | Few | All | Many | Med. | Few | All | Many | Med. | Few | All | Many | Med. | Few | All |
| Expert $E_1$ | 64.3 | 49.0 | 31.9 | 52.6 | 63.5 | 44.8 | 20.3 | 44.0 | **68.8** | 43.7 | 17.2 | 49.8 | **67.6** | 36.3 | 6.8 | 38.4 |
| Expert $E_2$ | 64.7 | 49.4 | 31.2 | 52.8 | 63.1 | 44.7 | 20.2 | 43.8 | 65.5 | 50.5 | 33.3 | **53.9** | 61.2 | 44.7 | 23.5 | **44.2** |
| Expert $E_3$ | 64.3 | 48.9 | 31.8 | 52.5 | 63.9 | 45.1 | 20.5 | 44.3 | 43.4 | 48.6 | **53.9** | 47.3 | 14.0 | 27.6 | **41.2** | 25.8 |
| Ensemble | 68.0 | 52.9 | 35.1 | 56.3 | 67.4 | 49.5 | 23.7 | 48.0 | 67.0 | 56.7 | 42.6 | **58.8** | 61.6 | 50.5 | 33.9 | **49.4** |

## 5.3 Superiority on Test-agnostic Long-tailed Recognition

In this subsection, we evaluate SADE on test-agnostic long-tailed recognition. The results on various test class distributions are reported in Table 5. Specifically, since Softmax seeks to simulate the long-tailed training distribution, it performs well on forward long-tailed test distributions. However, its performance on the uniform and backward long-tailed distributions is poor. In contrast, existing long-tailed methods show more balanced performance among classes, leading to better overall accuracy. However, the resulting models by these methods suffer from a simulation bias, *i.e.,* performing similarly among classes on various class distributions (c.f. Table 1). As a result, they cannot adapt to diverse test class distributions well. To handle this task, LADE assumes the test class distribution to be known and uses this information to adjust its predictions, leading to better performance on various test class distributions. However, since obtaining the actual test class distribution is difficult in real applications, the methods requiring such knowledge may be not applicable in practice. Moreover, in some specific cases like Forward-LT-3 and Backward-LT-3 distributions of iNaturalist 2018, the number of test samples on some classes becomes zero. In such cases, the test prior cannot be used in LADE, since adjusting logits with $\log 0$ results in biased predictions. In contrast, without relying on the knowledge of test class distributions, our SADE presents an innovative self-supervised strategy to deal with unknown class distributions, and obtains even better performance than LADE that uses the test class prior (c.f. Table 5). The promising results demonstrate the effectiveness and practicality of our method on test-agnostic long-tailed recognition. Note that the performance advantages of SADE become larger as the test data get more imbalanced. Due to the page limitation, the results on more datasets are reported in Appendix D.2.

## 5.4 Effectiveness of Skill-diverse Expert Learning

We next examine our skill-diverse expert learning strategy. The results are reported in Table 6, where RIDE [49] is a state-of-the-art ensemble-based method. RIDE trains each expert with cross-entropy independently and uses KL-Divergence to improve expert diversity. However, simply maximizing the divergence of expert predictions cannot learn visibly diverse experts (cf. Table 6). In contrast, the three experts learned by our strategy have significantly diverse expertise, excelling at many-shot classes, the uniform distribution (with higher overall performance), and few-shot classes, respectively. As a result, the increasing expert diversity leads to a non-trivial gain for the ensemble performance of SADE compared to RIDE. Moreover, consistent results on more datasets are reported in Appendix D.3, while the ablation studies of the expert learning strategy are provided in Appendix E.

Table 7: The expert weights learned by our self-supervised strategy on ImageNet-LT with various test class distributions. Our method learns suitable weights for various unknown distributions.

| Test Dist. | Expert $E_1$ ($w_1$) | Expert $E_2$ ($w_2$) | Expert $E_3$ ($w_3$) |
|---|---|---|---|
| Forward-LT-50 | 0.52 | 0.35 | 0.13 |
| Forward-LT-10 | 0.46 | 0.36 | 0.18 |
| Uniform | 0.33 | 0.33 | 0.34 |
| Backward-LT-10 | 0.21 | 0.29 | 0.50 |
| Backward-LT-50 | 0.17 | 0.27 | 0.56 |

Table 8: The performance improvement by our test-time self-supervised strategy on ImageNet-LT with various test class distributions.

| | ImageNet-LT | | | | | | | |
|---|---|---|---|---|---|---|---|---|
| Test Dist. | Ours w/o test-time strategy | | | | Ours w/ test-time strategy | | | |
| | Many | Med. | Few | All | Many | Med. | Few | All |
| Forward-LT-50 | 65.6 | 55.7 | 44.1 | 65.5 | 70.0 | 53.2 | 33.1 | 69.4 (+3.9) |
| Forward-LT-10 | 66.5 | 56.8 | 44.2 | 63.6 | 69.9 | 54.3 | 34.7 | 65.4 (+1.8) |
| Uniform | 67.0 | 56.7 | 42.6 | 58.8 | 66.5 | 57.0 | 43.5 | 58.8 (+0.0) |
| Backward-LT-10 | 65.0 | 57.6 | 43.1 | 53.1 | 60.9 | 57.5 | 50.1 | 54.5 (+1.4) |
| Backward-LT-50 | 69.1 | 57.0 | 42.9 | 49.8 | 60.7 | 56.2 | 50.7 | 53.1 (+3.3) |

Table 9: Comparison among different test-time training strategies for handling class distribution shifts on ImageNet-LT with various unknown test class distributions.

| Backbone | Test-time strategy | Forward | | | | | Uniform | Backward | | | | |
|---|---|---|---|---|---|---|---|---|---|---|---|---|
| | | 50 | 25 | 10 | 5 | 2 | 1 | 2 | 5 | 10 | 25 | 50 |
| | No test-time adaptation | 65.5 | 64.4 | 63.6 | 62.0 | 60.0 | 58.8 | 56.8 | 54.7 | 53.1 | 51.5 | 49.8 |
| | Test-time pseudo-labeling | 67.1 | 66.1 | 64.7 | 63.0 | 60.1 | 57.7 | 54.7 | 51.1 | 48.1 | 45.0 | 42.4 |
| SADE | Test class distribution estimation [29] | 69.1 | 66.6 | 63.7 | 60.5 | 56.5 | 53.3 | 49.9 | 45.6 | 42.7 | 39.5 | 36.8 |
| | Entropy minimization with Tent [46] | 68.0 | 67.0 | **65.6** | 62.8 | 60.5 | 58.6 | 56.0 | 53.2 | 50.6 | 48.1 | 45.7 |
| | Self-supervised expert aggregation (ours) | **69.4** | **67.4** | 65.4 | **63.0** | **60.6** | **58.8** | **57.1** | **55.5** | **54.5** | **53.7** | **53.1** |

## 5.5 Effectiveness of Test-time Self-supervised Aggregation

This subsection evaluates our test-time self-supervised aggregation strategy.

**Effectiveness in expert aggregation.** As shown in Table 7, our self-supervised strategy learns suitable expert weights for various unknown test class distributions. For forward long-tailed distributions, the weight of the forward expert $E_1$ is higher; while for backward long-tailed ones, the weight of the backward expert $E_3$ is relatively high. This enables our multi-expert model to boost the performance on dominant classes for unknown test distributions, leading to better ensemble performance (cf. Table 8), particularly as test data get more skewed. The results on more datasets are reported in Appendix D.4, while more ablation studies of our strategy are shown in Appendix F.

**Superiority over test-time training methods.** We then verify the superiority of our self-supervised strategy over existing test-time training approaches on various test class distributions. Specifically, we adopt three non-trivial baselines: (i) *Test-time pseudo-labeling* uses the multi-expert model to iteratively generate pseudo labels for unlabeled test data and uses them to fine-tune the model; (ii) *Test class distribution estimation* leverages BBSE [29] to estimate the test class distribution and uses it to pose-adjust model predictions; (iii) *Tent* [46] fine-tunes the batch normalization layers of models through entropy minimization on unlabeled test data. The results in Table 9 show that directly applying existing test-time training methods cannot handle well the class distribution shifts, particularly on the inversely long-tailed class distribution. In comparison, our self-supervised strategy is able to aggregate multiple experts appropriately for the unknown test class distribution (cf. Table 7), leading to promising performance gains on various test class distributions (cf. Table 9).

**Effectiveness on partial class distributions.**
Real-world test data may follow any type of class distribution, including partial class distributions (*i.e.,* not all of the classes appear in the test data). Motivated by this, we further evaluate SADE on three partial class distributions: only many-shot classes, only medium-shot classes, and only few-shot classes. The results in Table 10 demonstrate the effectiveness of SADE in tackling more complex test class distributions.

Table 10: The effectiveness of our self-supervised aggregation strategy in dealing with (unknown) partial test class distributions on ImageNet-LT.

| Method | ImageNet-LT | | |
|---|---|---|---|
| | Only many | Only medium | Only few |
| SADE w/o test-time strategy | 67.4 | 56.9 | 42.5 |
| SADE w/ test-time strategy | 71.0 | 57.2 | 53.6 |
| Accuracy gain | (+3.6) | (+0.3) | (+11.1) |

## 6 Conclusion

In this paper, we have explored a practical yet challenging task of *test-agnostic long-tailed recognition*, where the test class distribution is unknown and not necessarily uniform. To tackle this task, we present a novel approach, namely *Self-supervised Aggregation of Diverse Experts* (SADE), which consists of two innovative strategies, *i.e.,* skill-diverse expert learning and test-time self-supervised aggregation. We theoretically analyze our proposed method and also empirically show that SADE achieves new state-of-the-art performance on both vanilla and test-agnostic long-tailed recognition.

**Acknowledgments**

This work was partially supported by NUS ODPRT Grant R252-000-A81-133 and NUS Advanced Research and Technology Innovation Centre (ARTIC) Project Reference (ECT-RP2). We also gratefully appreciate the support of MindSpore, CANN (Compute Architecture for Neural Networks) and Ascend AI Processor used for this research.

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
