# Supplementary Material:
# Self-Supervised Aggregation of Diverse Experts for Test-Agnostic Long-Tailed Recognition

**Yifan Zhang**[1]    **Bryan Hooi**[1]    **Lanqing Hong**[2]    **Jiashi Feng**[3]
[1]National University of Singapore    [2]Huawei Noah's Ark Lab    [3]ByteDance
yifan.zhang@u.nus.edu, jshfeng@gmail.com

We organize the supplementary materials as follows:

• Appendix C: more details of experimental settings.

## A   Proofs for Theorem 1

*Proof.* We first recall several key notations and define some new notations. The random variables of model predictions and ground-truth labels are defined as $\hat{Y} \sim p(\hat{y})$ and $Y \sim p(y)$, respectively. The number of classes is denoted by $C$. Moreover, we further denote the test sample set of the class $k$ by $\mathcal{Z}_k$, in which the total number of samples in this class is denoted by $|\mathcal{Z}_k|$. Let $c_k = \frac{1}{|\mathcal{Z}_k|} \sum_{\hat{y} \in \mathcal{Z}_k} \hat{y}$ represent the hard mean of all predictions of samples from the class $k$, and let $\overset{c}{=}$ indicate equality up to a multiplicative and/or additive constant.

As shown in Eq. (4), the optimization objective of our test-time self-supervised aggregation method is to maximize $\mathcal{S} = \sum_{j=1}^{n_t} \hat{y}_j^1 \cdot \hat{y}_j^2$, where $n_t$ denotes the number of test samples. For convenience, we simplify the first data view to be the original data, so the objective function becomes $\sum_{j=1}^{n_t} \hat{y}_j \cdot \hat{y}_j^1$. Maximizing such an objective is equivalent to minimizing $\sum_{j=1}^{n_t} -\hat{y}_j \cdot \hat{y}_j^1$. Here, we assume the data augmentations are strong enough to generate representative data views that can simulate the test data from the same class. In this sense, the new data view can be regarded as an independent sample from the same class. Following this, we analyze our method by connecting $-\hat{y}_j \cdot \hat{y}_j^1$ to $\sum_{\hat{y}_j \in \mathcal{Z}_k} \|\hat{y}_j - c_k\|^2$,

which is similar to the tightness term in the center loss [18]:

$$\sum_{\hat{y}_j,\hat{y}_j^1\in\mathcal{Z}_k} -\hat{y}_j\cdot\hat{y}_j^1 \overset{c}{=} \frac{1}{|\mathcal{Z}_k|}\sum_{\hat{y}_j,\hat{y}_j^1\in\mathcal{Z}_k} -\hat{y}_j\cdot\hat{y}_j^1 \overset{c}{=} \frac{1}{|\mathcal{Z}_k|}\sum_{\hat{y}_j,\hat{y}_j^1\in\mathcal{Z}_k} \|\hat{y}_j\|^2 - \hat{y}_j\cdot\hat{y}_j^1$$

$$= \sum_{\hat{y}_j\in\mathcal{Z}_k}\|\hat{y}_j\|^2 - \frac{1}{|\mathcal{Z}_k|}\sum_{\hat{y}_j\in\mathcal{Z}_k}\sum_{\hat{y}_j^1\in\mathcal{Z}_k}\hat{y}_j\cdot\hat{y}_j^1$$

$$= \sum_{\hat{y}_j\in\mathcal{Z}_k}\|\hat{y}_j\|^2 - 2\frac{1}{|\mathcal{Z}_k|}\sum_{\hat{y}_j\in\mathcal{Z}_k}\sum_{\hat{y}_j^1\in\mathcal{Z}_k}\hat{y}_j\cdot\hat{y}_j^1 + \frac{1}{|\mathcal{Z}_k|}\sum_{\hat{y}_j\in\mathcal{Z}_k}\sum_{\hat{y}_j^1\in\mathcal{Z}_k}\hat{y}_j\cdot\hat{y}_j^1$$

$$= \sum_{\hat{y}_j\in\mathcal{Z}_k}\|\hat{y}_j\|^2 - 2\hat{y}_j\cdot c_k + \|c_k\|^2$$

$$= \sum_{\hat{y}_j\in\mathcal{Z}_k}\|\hat{y}_j - c_k\|^2,$$

where we use the property of the normalized predictions, *i.e.*, $\|\hat{y}_j\|^2 = \|\hat{y}_j^1\|^2 = 1$, and the definition of the class hard mean $c_k = \frac{1}{|\mathcal{Z}_k|}\sum_{\hat{y}\in\mathcal{Z}_k}\hat{y}$.

By summing over all classes $k$, we obtain:

$$\sum_{j=1}^{n_t}-\hat{y}_j\cdot\hat{y}_j^1 \overset{c}{=} \sum_{j=1}^{n_t}\|\hat{y}_j - c_{y_i}\|^2.$$

Based on this equation, following [1, 21], we can interpret $\sum_{j=1}^{n_t}-\hat{y}_j\cdot\hat{y}_j^1$ as a conditional cross-entropy between $\hat{Y}$ and another random variable $\bar{Y}$, whose conditional distribution given $Y$ is a standard Gaussian centered around $c_Y:\bar{Y}|Y\sim\mathcal{N}(c_y, i)$:

$$\sum_{j=1}^{n_t}-\hat{y}_j\cdot\hat{y}_j^1 \overset{c}{=} \mathcal{H}(\hat{Y};\bar{Y}|Y) = \mathcal{H}(\hat{Y}|Y)+\mathcal{D}_{KL}(\hat{Y}\|\bar{Y}|Y).$$

Hence, we know that $\sum_{j=1}^{n_t}-\hat{y}_j\cdot\hat{y}_j^1$ is an upper bound on the conditional entropy of predictions $\hat{Y}$ given labels $Y$:

$$\sum_{j=1}^{n_t}-\hat{y}_j\cdot\hat{y}_j^1 \overset{c}{\geq} \mathcal{H}(\hat{Y}|Y),$$

where the symbol $\overset{c}{\geq}$ represents "larger than" up to a multiplicative and/or an additive constant. Moreover, when $\hat{Y}|Y\sim\mathcal{N}(c_y, i)$, the bound is tight. As a result, minimizing $\sum_{j=1}^{n_t}-\hat{y}_j\cdot\hat{y}_j^1$ is equivalent to minimizing $\mathcal{H}(\hat{Y}|Y)$:

$$\sum_{j=1}^{n_t}-\hat{y}_j\cdot\hat{y}_j^1 \propto \mathcal{H}(\hat{Y}|Y).$$

Meanwhile, the mutual information between predictions $\hat{Y}$ and labels $Y$ can be represented by:

$$\mathcal{I}(\hat{Y};Y) = \mathcal{H}(\hat{Y}) - \mathcal{H}(\hat{Y}|Y).$$

Combining the above two equations, we have:

$$\sum_{j=1}^{n_t}-\hat{y}_j\cdot\hat{y}_j^1 \propto -\mathcal{I}(\hat{Y};Y) + \mathcal{H}(\hat{Y}).$$

Since $\mathcal{S} = \sum_{j=1}^{n_t}\hat{y}_j\cdot\hat{y}_j^1$, we obtain:

$$\mathcal{S} \propto \mathcal{I}(\hat{Y};Y) - \mathcal{H}(\hat{Y}),$$

which concludes the proof for Theorem 1. $\square$

# B  Pseudo-code

This appendix provides the pseudo-code[1] of SADE, which consists of skill-diverse expert learning and test-time self-supervised aggregation. Here, the skill-diverse expert learning strategy is summarized in Algorithm 1. For simplicity, we depict the pseudo-code based on batch size 1, but we conduct batch gradient descent in practice.

---

**Algorithm 1** Skill-diverse Expert Learning

---

**Require:** Epochs $T$; Hyper-parameters $\lambda$ for $\mathcal{L}_{inv}$
**Initialize:** Network backbone $f_\theta$; Experts $E_1, E_2, E_3$
 1: **for** e=1,...,$T$ **do**
 2:     **for** $x \in \mathcal{D}_s$ **do**  // batch sampling in practice
 3:         Obtain logits $v_1$ based on $f_\theta$ and $E_1$;
 4:         Obtain logits $v_2$ based on $f_\theta$ and $E_2$;
 5:         Obtain logits $v_3$ based on $f_\theta$ and $E_3$;
 6:         Compute loss $\mathcal{L}_{ce}$ with $v_1$ for Expert $E_1$;  // Eq. (1)
 7:         Compute loss $\mathcal{L}_{bal}$ with $v_2$ for Expert $E_2$;  // Eq. (2)
 8:         Compute loss $\mathcal{L}_{inv}$ with $v_3$ for Expert $E_3$;  // Eq. (3)
 9:         Train the model with $\mathcal{L}_{ce} + \mathcal{L}_{bal} + \mathcal{L}_{inv}$.
10:     **end for**
11: **end for**
**Output:** The trained model $\{f_\theta, E_1, E_2, E_3\}$

---

After training the multiple skill-diverse experts with Algorithm 1, the final prediction of the multi-expert model for vanilla long-tailed recognition is the arithmetic mean of the prediction logits of these experts, followed by a softmax function.

When it comes to test-agnostic long-tailed recognition, we need to aggregate these skill-diverse experts to handle the unknown test class distribution based on Algorithm 2. Here, to avoid the learned weights of some weak experts becoming zero, we give a stopping condition in Algorithm 2: if the weight for one expert is less than 0.05, we stop test-time training. Retaining a small amount of weight for each expert is sufficient to ensure the effect of ensemble learning.

---

**Algorithm 2** Test-time Self-supervised Aggregation

---

**Require:** Epochs $T'$; The trained backbone $f_\theta$; The trained experts $E_1, E_2, E_3$
**Initialize:** Expert aggregation weights $w$ // uniform initialization
 1: **for** e=1,...,$T'$ **do**
 2:     **for** $x \in \mathcal{D}_t$ **do**  // batch sampling in practice
 3:         Draw two data augmentation functions $t \sim \mathcal{T}$, $t' \sim \mathcal{T}$;
 4:         Generate data views $x^1 = t(x)$, $x^2 = t'(x)$;
 5:         Obtain logits $v_1^1, v_2^1, v_3^1$ for the view $x^1$;
 6:         Obtain logits $v_1^2, v_2^2, v_3^2$ for the view $x^2$;
 7:         Normalize expert weights $w$ via softmax function;
 8:         Conduct predictions $\hat{y}^1, \hat{y}^2$ based on $\hat{y} = wv$;
 9:         Compute prediction stability $\mathcal{S}$;  // Eq. (4)
10:         Maximize $\mathcal{S}$ to update $w$;
11:     **end for**
12:     If $w_i \leq 0.05$ for any $w_i \in w$, then stop training.
13: **end for**
**Output:** Expert aggregation weights $w$

---

Note that, in test-agnostic long-tailed recognition, each model is only trained once on long-tailed training data and then directly evaluated on multiple test sets. Our test-time self-supervised strategy adapts the trained multi-expert model using only unlabeled test data during testing.

---

[1]The source code is provided at https://github.com/Vanint/SADE-AgnosticLT.

## C  More Experimental Settings

In this appendix, we provide more details on experimental settings.

### C.1  Benchmark Datasets

We use four benchmark datasets [22] (*i.e.,* ImageNet-LT [12], CIFAR100-LT [2], Places-LT [12], and iNaturalist 2018 [15]) to simulate real-world long-tailed class distributions. These datasets suffer from severe class imbalance [10, 24].Their data statistics are summarized in Table 1, where CIFAR100-LT has three variants with different imbalance ratios. The imbalance ratio is defined as $\max n_j/\min n_j$, where $n_j$ denotes the data number of class $j$.

Table 1: Statistics of datasets.

| Dataset | # classes | # training data | # test data | imbalance ratio |
|---|---|---|---|---|
| ImageNet-LT [12] | 1,000 | 115,846 | 50,000 | 256 |
| CIFAR100-LT [2] | 100 | 50,000 | 10,000 | {10,50,100} |
| Places-LT [12] | 365 | 62,500 | 36,500 | 996 |
| iNaturalist 2018 [15] | 8,142 | 437,513 | 24,426 | 500 |

### C.2  Construction of Test-agnostic Long-tailed Datasets

Following LADE [8], we construct three kinds of test class distributions, *i.e.,* the uniform distribution, forward long-tailed distributions and backward long-tailed distributions. In the backward ones, the long-tailed class order is flipped. Here, the forward and backward long-tailed test distributions contain multiple different imbalance ratios, *i.e.,* $\rho \in \{2, 5, 10, 25, 50\}$. Note that LADE [8] only constructed multiple distribution-agnostic test datasets for ImageNet-LT; while in this study, we use the same way to construct distribution-agnostic test datasets for the remaining benchmark datasets, *i.e.,* CIFAR100-LT, Places-LT and iNaturalist 2018, as illustrated below.

Considering the long-tailed training classes are sorted in a decreasing order, the various test datasets are constructed as follows: (1) Forward long-tailed distribution: the number of the $j$-th class is $n_j = N \cdot \rho^{(j-1)/C}$, where $N$ indicates the sample number per class in the original uniform test dataset and $C$ is the number of classes. (2) Backward long-tailed distribution: the number of the $j$-th class is $n_j = N \cdot \rho^{(C-j)/C}$. In the backward long-tailed distributions, the order of the long tail on classes is flipped, so the distribution shift between training and test data is large, especially when the imbalance ratio gets higher.

For ImageNet-LT, CIFAR100-LT and Places-LT, since there are enough test samples per class, we follow the setting in LADE [8] and construct the imbalance ratio set by $\rho \in \{2, 5, 10, 25, 50\}$. For iNaturalist 2018, since each class only contains three test samples, we adjust the imbalance ratio set to $\rho \in \{2, 3\}$. Note that when we set $\rho = 3$, there are some classes in iNaturalist 2018 containing no test sample. All these constructed distribution-agnostic long-tailed datasets will be publicly available along with our code.

### C.3  More Implementation Details of Our Method

We implement our method in PyTorch. Following [8, 17], we use ResNeXt-50 for ImageNet-LT, ResNet-32 for CIFAR100-LT, ResNet-152 for Places-LT and ResNet-50 for iNaturalist 2018 as backbones, respectively. Moreover, we adopt the cosine classifier for prediction on all datasets.

Although we have depicted the skill-diverse multi-expert framework in Section 4.1, we give more details about it here. Without loss of generality, we take ResNet [7] as an example to illustrate the multi-expert model. Since the shallow layers extract more general features and deeper layers extract more task-specific features [20], the three-expert model uses the first two stages of ResNet as the expert-shared backbone, while the later stages of ResNet and the fully-connected layer constitute independent components of each expert. To be more specific, the number of convolutional filters in each expert is reduced by 1/4, since by sharing the backbone and using fewer filters in each expert [17, 26], the computational complexity of the model is reduced compared to the model with independent experts. The final prediction is the arithmetic mean of the prediction logits of these experts, followed by a softmax function.

In the training phase, the data augmentations are the same as previous long-tailed studies [8, 11]. If not specified, we use the SGD optimizer with the momentum of 0.9 and set the initial learning rate as 0.1 with linear decay. More specifically, for ImageNet-LT, we train models for 180 epochs with batch size 64 and a learning rate of 0.025 (cosine decay). For CIFAR100-LT, the training epoch is 200 and the batch size is 128. For Places-LT, following [12], we use ImageNet pre-trained ResNet-152 as the backbone, while the batch size is set to 128 and the training epoch is 30. Besides, the learning rate is 0.01 for the classifier and 0.001 for all other layers. For iNaturalist 2018, we set the training epoch to 200, the batch size to 512 and the learning rate to 0.2. In our inverse softmax loss, we set $\lambda=2$ for ImageNet-LT and CIFAR100-LT, and $\lambda=1$ for the remaining datasets.

In the test-time training, we use the same augmentations as MoCo v2 [3] to generate different data views, *i.e.,* random resized crop, color jitter, gray scale, Gaussian blur and horizontal flip. If not specified, we train the aggregation weights for 5 epochs with the batch size 128, where we adopt the same optimizer and learning rate as the training phase.

More detailed statistics of network architectures and hyper-parameters are reported in Table 2. Based on these hyper-parameters, we conduct experiments on 1 TITAN RTX 2080 GPU for CIFAR100-LT, 4 GPUs for iNaturalist18, and 2 GPUs for ImageNet-LT and Places-LT, respectively.

Table 2: Statistics of the used network architectures and hyper-parameters in our study.

| Items | ImageNet-LT | CIFAR100LT | Places-LT | iNarutalist 2018 |
|---|---|---|---|---|
| Network Architectures | | | | |
| network backbone | ResNeXt-50 | ResNet-32 | ResNet-152 | ResNet-50 |
| classifier | cosine classifier | | | |
| Training Phase | | | | |
| epochs | 180 | 200 | 30 | 200 |
| batch size | 64 | 128 | 128 | 512 |
| learning rate (lr) | 0.025 | 0.1 | 0.01 | 0.2 |
| lr schedule | cosine decay | linear decay | | |
| $\lambda$ in inverse softmax loss | 2 | | 1 | |
| weight decay factor | $5 \times 10^{-4}$ | $5 \times 10^{-4}$ | $4 \times 10^{-4}$ | $2 \times 10^{-4}$ |
| momentum factor | 0.9 | | | |
| optimizer | SGD optimizer with nesterov | | | |
| Test-time Training | | | | |
| epochs | 5 | | | |
| batch size | 128 | | | |
| learning rate (lr) | 0.025 | 0.1 | 0.01 | 0.1 |

## C.4 Discussions on Evaluation Metric

As mentioned in Section 5.1, we follow LADE [8] and use micro accuracy to evaluate model performance on test-agnostic long-tailed recognition. In this appendix, we explain why micro accuracy is a better metric than macro accuracy when the test dataset exhibits a non-uniform class distribution. For instance, in the test scenario with a backward long-tailed class distribution, the tail classes are more frequently encountered than the head classes, and thus should have larger weights in evaluation. However, simply using macro accuracy treats all the categories equally and cannot differentiate classes of different frequencies.

For example, one may train a recognition model for autonomous cars based on the training data collected from city areas, where pedestrians are majority classes and stone obstacles are minority classes. Assume the model accuracy is 60% on pedestrians and 40% on stones. If deploying the model to city areas, where pedestrians/stones are assumed to have 500/50 test data, then the macro accuracy is 50% and the micro accuracy is $\frac{500 \times 0.6 + 50 \times 0.4}{500 + 50} \approx 58\%$. In contrast, when deploying the model to mountain areas, the pedestrians become the minority, while stones become the majority. Assuming the test data numbers are changed to 50/500 on pedestrians/stones, the micro accuracy is adjusted to $\frac{50 \times 0.6 + 500 \times 0.4}{50 + 500} \approx 42\%$, but the macro accuracy is still 50%. In this case, macro accuracy is less informative than micro accuracy for measuring model performance. Therefore, micro accuracy is a better metric to evaluate the performance of test-agnostic long-tailed recognition.

# D  More Empirical Results

## D.1  More Results on Vanilla Long-tailed Recognition

**Accuracy on class subsets**  In the main paper, we have provided the average performance over all classes on the uniform test class distribution. In this appendix, we further report the accuracy regarding various class subsets (c.f. Table 3), making the results more complete.

Table 3: Top-1 accuracy of long-tailed recognition methods on the uniform test distribution.

| Method | ImageNet-LT | | | | CIFAR100-LT(IR10) | | | | CIFAR100-LT(IR50) | | | |
|---|---|---|---|---|---|---|---|---|---|---|---|---|
| | Many | Med. | Few | All | Many | Med. | Few | All | Many | Med. | Few | All |
| Softmax | **68.1** | 41.5 | 14.0 | 48.0 | **66.0** | 42.7 | - | 59.1 | **66.8** | 37.4 | 15.5 | 45.6 |
| Causal [14] | 64.1 | 45.8 | 27.2 | 50.3 | 63.3 | 49.9 | - | 59.4 | 62.9 | 44.9 | 26.2 | 48.8 |
| Balanced Softmax [9] | 64.1 | 48.2 | 33.4 | 52.3 | 63.4 | 55.7 | - | 61.0 | 62.1 | 45.6 | 36.7 | 50.9 |
| MiSLAS [25] | 62.0 | 49.1 | 32.8 | 51.4 | 64.9 | 56.6 | - | 62.5 | 61.8 | 48.9 | 33.9 | 51.5 |
| LADE [8] | 64.4 | 47.7 | 34.3 | 52.3 | 63.8 | 56.0 | - | 61.6 | 60.2 | 46.2 | 35.6 | 50.1 |
| RIDE [17] | 68.0 | 52.9 | 35.1 | 56.3 | 65.7 | 53.3 | - | 61.8 | 66.6 | 46.2 | 30.3 | 51.7 |
| SADE (ours) | 66.5 | **57.0** | **43.5** | **58.8** | 65.8 | **58.8** | - | **63.6** | 61.5 | **50.2** | **45.0** | **53.9** |

| Method | CIFAR100-LT(IR100) | | | | Places-LT | | | | iNaturalist 2018 | | | |
|---|---|---|---|---|---|---|---|---|---|---|---|---|
| | Many | Med. | Few | All | Many | Med. | Few | All | Many | Med. | Few | All |
| Softmax | **68.6** | 41.1 | 9.6 | 41.4 | **46.2** | 27.5 | 12.7 | 31.4 | **74.7** | 66.3 | 60.0 | 64.7 |
| Causal [14] | 64.1 | 46.8 | 19.9 | 45.0 | 23.8 | 35.7 | **39.8** | 32.2 | 71.0 | 66.7 | 59.7 | 64.4 |
| Balanced Softmax [9] | 59.5 | 45.4 | **30.7** | 46.1 | 42.6 | 39.8 | 32.7 | 39.4 | 70.9 | 70.7 | 70.4 | 70.6 |
| MiSLAS [25] | 60.4 | **49.6** | 26.6 | 46.8 | 41.6 | 39.3 | 27.5 | 37.6 | 71.7 | 71.5 | 69.7 | 70.7 |
| LADE [8] | 58.7 | 45.8 | 29.8 | 45.6 | 42.6 | 39.4 | 32.3 | 39.2 | 68.9 | 68.7 | 70.2 | 69.3 |
| RIDE [17] | 67.4 | 49.5 | 23.7 | 48.0 | 43.1 | 41.0 | 33.0 | 40.3 | 71.5 | 70.0 | 71.6 | 71.8 |
| SADE (ours) | 65.4 | 49.3 | 29.3 | **49.8** | 40.4 | **43.2** | 36.8 | **40.9** | 74.5 | **72.5** | **73.0** | **72.9** |

**Results on stonger data augmentations**  Inspired by PaCo [5], we further evaluate SADE training with stronger data augmentation (*i.e.,* RandAugment [4]) for 400 epochs. The results in Table 4 further demonstrate the state-of-the-art performance of SADE.

Table 4: Accuracy of long-tailed methods with stronger augmentations, where the test class distribution is uniform. Here, * denotes training with RandAugment [4] for 400 epochs. The baseline results are directly copied from the work [5].

| Methods | ImageNet-LT | CIFAR100-LT(IR10) | CIFAR100-LT(IR50) | CIFAR100-LT(IR100) | Places-LT | iNaturalist 2018 |
|---|---|---|---|---|---|---|
| PaCo* [5] | 58.2 | 64.2 | 56.0 | 52.0 | 41.2 | 73.2 |
| SADE* (ours) | **61.2** | **65.3** | **57.3** | **52.2** | **41.3** | **74.5** |

**Results on more neural architectures**  In addition to using the common practice of backbones as previous long-tailed studies [8, 17], we further evaluate SADE on more neural architectures. The results in Table 5 demonstrate that SADE is able to train different network backbones well.

Table 5: Accuracy of SADE with various network architectures. Here, * denotes training with RandAugment [4] for 400 epochs.

| Backbone | Methods | ImageNet-LT | | | | Backbone | Methods | iNaturalist 2018 | | | |
|---|---|---|---|---|---|---|---|---|---|---|---|
| | | Many | Med. | Few | All | | | Many | Med. | Few | All |
| ResNeXt-50 | SADE | 66.5 | 57.0 | 43.5 | 58.8 | ResNet-50 | SADE | 74.5 | 72.5 | 73.0 | 72.9 |
| | SADE* | 67.3 | 60.4 | 46.4 | 61.2 | | SADE* | 75.5 | 73.7 | 75.1 | 74.5 |
| ResNeXt-101 | SADE | 66.8 | 57.5 | 43.1 | 59.1 | ResNet-152 | SADE | 76.2 | 64.3 | 65.1 | 74.8 |
| | SADE* | 68.1 | 60.5 | 45.5 | 61.4 | | SADE* | **78.3** | **77.0** | **76.7** | **77.0** |
| ResNeXt-152 | SADE | 67.2 | 57.4 | 43.5 | 59.3 | | | | | | |
| | SADE* | **68.6** | **61.2** | **47.0** | **62.1** | | | | | | |

**Results on more datasets**    We also conduct experiments on CIFAR10-LT with imbalance ratios of 10 and 100. Promising results in Table 6 further demonstrate the effectiveness and superiority of our proposed method.

Table 6: Accuracy on CIFAR10-LT, where the test class distribution is uniform. Most results are directly copied from the work [25].

| Imbalance Ratio | Softmax | BBN | MiSLAS | RIDE | SADE (ours) |
|:---:|:---:|:---:|:---:|:---:|:---:|
| 10 | 86.4 | 88.4 | 90.0 | 89.7 | **90.8** |
| 100 | 70.4 | 79.9 | 82.1 | 81.6 | **83.8** |

## D.2 More Results on Test-agnostic Long-tailed Recognition

In the main paper, we have provided the overall performance on four benchmark datasets with various test class distributions. In this appendix, we further verify the effectiveness of our method on more dataset settings (*i.e.,* CIFAR100-IR10 and CIFAR100-IR50), as shown in Table 7.

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

Furthermore, we plot the results of all methods under these benchmark datasets with various test class distributions in Figure 1. To be specific, Softmax only performs well on highly-imbalanced forward long-tailed class distributions. Existing long-tailed baselines outperform Softmax, but they cannot handle backward test class distributions well. In contrast, our method consistently outperforms baselines on all benchmark datasets, particularly on the backward long-tailed test distributions with a relatively large imbalance ratio.

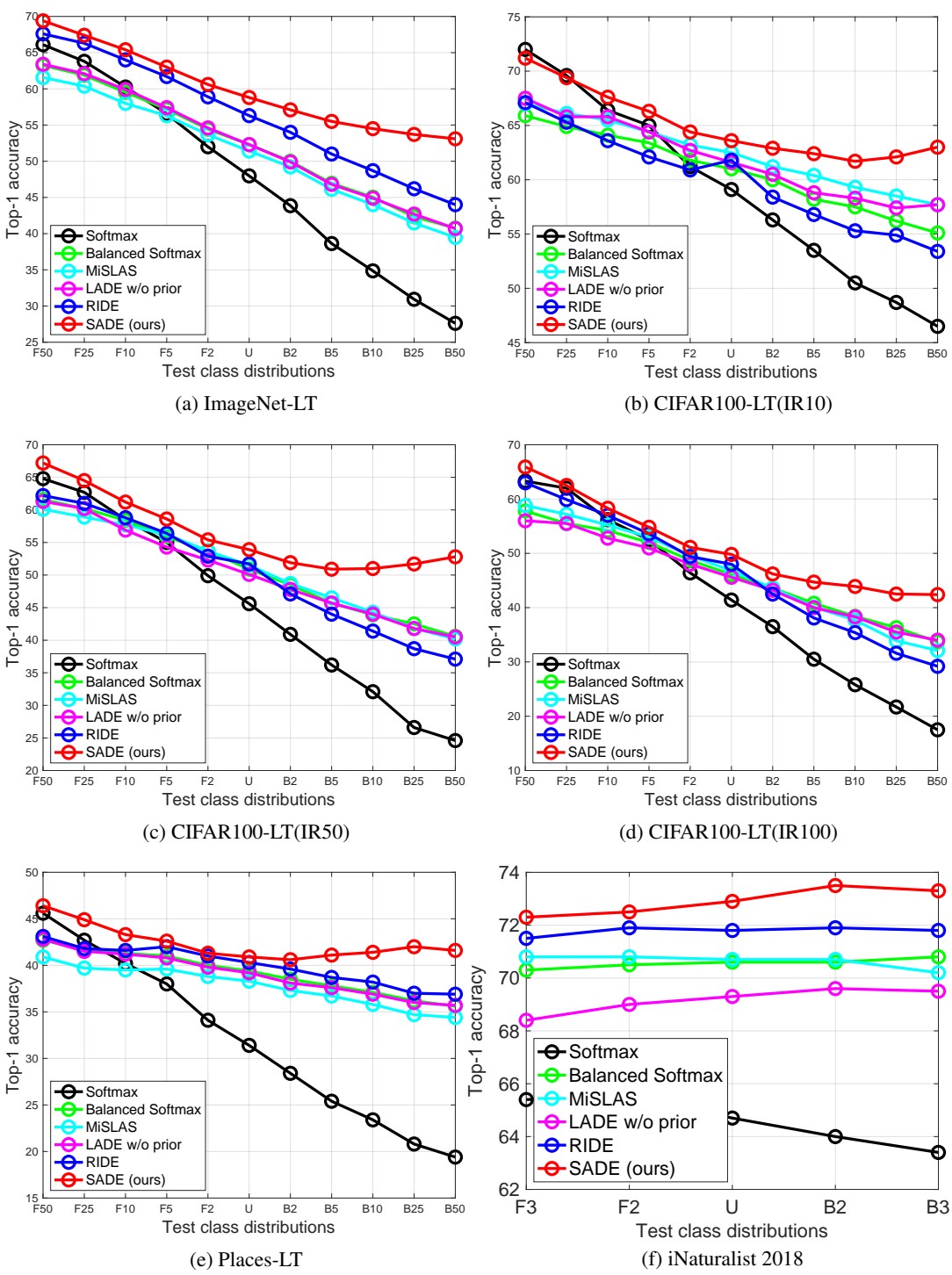

Figure 1: Performance visualizations on various unknown test class distributions, where "F" indicates the forward long-tailed distributions as training data, "B" indicates the backward long-tailed distributions to the training data, and "U" denotes the uniform distribution.

### D.3 More Results on Skill-diverse Expert Learning

This appendix further evaluates the skill-diverse expert learning strategy on CIFAR100-LT, Places-LT and iNaturalist 2018 datasets. We report the results in Table 8, from which we draw the following observations. RIDE [17] is one of the state-of-the-art ensemble-based long-tailed methods, which tries to learn diverse distribution-aware experts by maximizing the divergence among expert predictions. However, such a method cannot learn sufficiently diverse experts. As shown in Table 8, the three experts in RIDE perform very similarly on various groups of classes under all benchmark datasets, and each expert has similar overall performance on each dataset. Such results demonstrate that simply maximizing the KL divergence of different experts' predictions is not sufficient to learn visibly diverse distribution-aware experts.

In contrast, our proposed method learns the skill-diverse experts by directly training each expert with their customized expertise-guided objective functions, respectively. To be specific, the forward expert $E_1$ seeks to learn the long-tailed training distribution, so we directly train it with the cross-entropy loss. For the uniform expert $E_2$, we use the balanced softmax loss to simulate the uniform test distribution. For the backward expert $E_3$, we design a novel inverse softmax loss to train the expert, so that it simulates the inversely long-tailed class distribution. Table 8 shows that the three experts trained by our method are visibly diverse and skilled in handling different class distributions. Specifically, the forward expert is skilled in many-shot classes, the uniform expert is more balanced with higher overall performance, and the backward expert is good at few-shot classes. Because of such a novel design that enhances expert diversity, our method achieves more promising ensemble performance compared to RIDE.

Table 8: Performance of each expert on the uniform test distribution. Here, the training imbalance ratio of CIFAR100-LT is 100. The results show that our proposed method learns more skill-diverse experts, leading to better performance of ensemble aggregation.

| | | RIDE [17] | | | | | | | | | | | | | | |
|---|---|---|---|---|---|---|---|---|---|---|---|---|---|---|---|---|
| Model | ImageNet-LT | | | | CIFAR100-LT | | | | Places-LT | | | | iNaturalist 2018 | | | |
| | Many | Med. | Few | All | Many | Med. | Few | All | Many | Med. | Few | All | Many | Med. | Few | All |
| Expert $E_1$ | 64.3 | 49.0 | 31.9 | 52.6 | 63.5 | 44.8 | 20.3 | 44.0 | 41.3 | 40.8 | 33.2 | 40.1 | 66.6 | 67.1 | 66.5 | 66.8 |
| Expert $E_2$ | 64.7 | 49.4 | 31.2 | 52.8 | 63.1 | 44.7 | 20.2 | 43.8 | 43.0 | 40.9 | 33.6 | 40.3 | 66.1 | 67.1 | 66.6 | 66.8 |
| Expert $E_3$ | 64.3 | 48.9 | 31.8 | 52.5 | 63.9 | 45.1 | 20.5 | 44.3 | 42.8 | 41.0 | 33.5 | 40.2 | 65.3 | 67.3 | 66.5 | 66.7 |
| Ensemble | 68.0 | 52.9 | 35.1 | 56.3 | 67.4 | 49.5 | 23.7 | 48.0 | 43.2 | 41.1 | 33.5 | 40.3 | 71.5 | 72.0 | 71.6 | 71.8 |
| | | SADE (ours) | | | | | | | | | | | | | | |
| Model | ImageNet-LT | | | | CIFAR100-LT | | | | Places-LT | | | | iNaturalist 2018 | | | |
| | Many | Med. | Few | All | Many | Med. | Few | All | Many | Med. | Few | All | Many | Med. | Few | All |
| Expert $E_1$ | **68.8** | 43.7 | 17.2 | 49.8 | **67.6** | 36.3 | 6.8 | 38.4 | **47.6** | 27.1 | 10.3 | 31.2 | **76.0** | 67.1 | 59.3 | 66.0 |
| Expert $E_2$ | 65.5 | 50.5 | 33.3 | 53.9 | 61.2 | 44.7 | 23.5 | 44.2 | 42.6 | 42.3 | 32.3 | 40.5 | 69.2 | 70.7 | 69.8 | 70.2 |
| Expert $E_3$ | 43.4 | 48.6 | **53.9** | 47.3 | 14.0 | 27.6 | **41.2** | 25.8 | 22.6 | 37.2 | **45.6** | 33.6 | 55.6 | 61.5 | **72.1** | 65.1 |
| Ensemble | 67.0 | 56.7 | 42.6 | **58.8** | 61.6 | 50.5 | 33.9 | **49.4** | 40.4 | 43.2 | 36.8 | **40.9** | 74.4 | 72.5 | 73.1 | **72.9** |

## D.4 More Results on Test-time Self-supervised Aggregation

This appendix provides more results to examine the effectiveness of our test-time self-supervised aggregation strategy. We report results in Table 9, from which we draw several observations.

First of all, our method is able to learn suitable expert aggregation weights for test-agnostic class distributions, without relying on the true test class distribution. For the forward long-tailed test distribution, where the test data number of many-shot classes is more than that of medium-shot and few-shot classes, our method learns a higher weight for the forward expert $E_1$ who is skilled in many-shot classes, and learns relatively low weights for the expert $E_2$ and expert $E_3$ who are good at medium-shot and few-shot classes. Meanwhile, for the uniform test class distribution where all classes have the same number of test samples, our test-time expert aggregation strategy learns relatively balanced weights for the three experts. For example, on the uniform ImageNet-LT test data, the learned weights by our strategy are 0.33, 0.33 and 0.34 for the three experts, respectively. In addition, for the backward long-tailed test distributions, our method learns a higher weight for the backward expert $E_3$ and a relatively low weight for the forward expert $E_1$. Note that when the class imbalance ratio becomes larger, our method is able to learn more diverse expert weights adaptively for fitting the actual test class distributions.

Such results not only demonstrate the effectiveness of our proposed strategy, but also verify the theoretical analysis that our method can simulate the unknown test class distribution. To our best knowledge, such an ability is quite promising, since it is difficult to know the true test class distributions in real-world application. Therefore, our method opens the opportunity for tackling unknown class distribution shifts at test time, and can serve as a better candidate to handle real-world long-tailed learning applications.

Table 9: The learned aggregation weights by our test-time self-supervised aggregation strategy on different test class distributions of ImageNet-LT, CIFAR100-LT, Places-LT and iNaturalist 2018. The results show that our self-supervised strategy is able to learn suitable expert weights for various unknown test class distributions.

| Test Dist. | ImageNet-LT | | | CIFAR100-LT(IR10) | | | CIFAR100-LT(IR50) | | |
|---|---|---|---|---|---|---|---|---|---|
| | E1 ($w_1$) | E2 ($w_2$) | E3 ($w_3$) | E1 ($w_1$) | E2 ($w_2$) | E3 ($w_3$) | E1 ($w_1$) | E2 ($w_2$) | E3 ($w_3$) |
| Forward-LT-50 | 0.52 | 0.35 | 0.13 | 0.53 | 0.38 | 0.09 | 0.55 | 0.38 | 0.07 |
| Forward-LT-25 | 0.50 | 0.35 | 0.15 | 0.52 | 0.37 | 0.11 | 0.54 | 0.38 | 0.08 |
| Forward-LT-10 | 0.46 | 0.36 | 0.18 | 0.47 | 0.36 | 0.17 | 0.52 | 0.37 | 0.11 |
| Forward-LT-5 | 0.43 | 0.34 | 0.23 | 0.46 | 0.34 | 0.20 | 0.50 | 0.36 | 0.14 |
| Forward-LT-2 | 0.37 | 0.35 | 0.28 | 0.39 | 0.37 | 0.24 | 0.39 | 0.38 | 0.23 |
| Uniform | 0.33 | 0.33 | 0.34 | 0.38 | 0.32 | 0.3 | 0.35 | 0.33 | 0.33 |
| Backward-LT-2 | 0.29 | 0.31 | 0.40 | 0.35 | 0.33 | 0.31 | 0.30 | 0.30 | 0.40 |
| Backward-LT-5 | 0.24 | 0.31 | 0.45 | 0.31 | 0.32 | 0.37 | 0.21 | 0.29 | 0.50 |
| Backward-LT-10 | 0.21 | 0.29 | 0.50 | 0.26 | 0.32 | 0.42 | 0.20 | 0.29 | 0.51 |
| Backward-LT-25 | 0.18 | 0.29 | 0.53 | 0.24 | 0.30 | 0.46 | 0.18 | 0.27 | 0.55 |
| Backward-LT-50 | 0.17 | 0.27 | 0.56 | 0.23 | 0.28 | 0.49 | 0.14 | 0.24 | 0.62 |

| Test Dist. | CIFAR100-LT(IR100) | | | Places-LT | | | iNaturalist 2018 | | |
|---|---|---|---|---|---|---|---|---|---|
| | E1 ($w_1$) | E2 ($w_2$) | E3 ($w_3$) | E1 ($w_1$) | E2 ($w_2$) | E3 ($w_3$) | E1 ($w_1$) | E2 ($w_2$) | E3 ($w_3$) |
| Forward-LT-50 | 0.56 | 0.38 | 0.06 | 0.50 | 0.20 | 0.20 | - | - | - |
| Forward-LT-25 | 0.55 | 0.38 | 0.07 | 0.50 | 0.20 | 0.20 | - | - | - |
| Forward-LT-10 | 0.52 | 0.39 | 0.09 | 0.50 | 0.20 | 0.20 | - | - | - |
| Forward-LT-5 | 0.51 | 0.37 | 0.12 | 0.46 | 0.32 | 0.22 | - | - | - |
| Forward-LT-2 | 0.49 | 0.35 | 0.16 | 0.40 | 0.34 | 0.26 | 0.41 | 0.34 | 0.25 |
| Uniform | 0.40 | 0.35 | 0.24 | 0.25 | 0.34 | 0.41 | 0.33 | 0.33 | 0.34 |
| Backward-LT-2 | 0.33 | 0.31 | 0.36 | 0.18 | 0.30 | 0.52 | 0.28 | 0.32 | 0.40 |
| Backward-LT-5 | 0.28 | 0.30 | 0.42 | 0.17 | 0.28 | 0.55 | - | - | - |
| Backward-LT-10 | 0.23 | 0.28 | 0.49 | 0.17 | 0.27 | 0.56 | - | - | - |
| Backward-LT-25 | 0.21 | 0.26 | 0.53 | 0.17 | 0.27 | 0.56 | - | - | - |
| Backward-LT-50 | 0.16 | 0.28 | 0.56 | 0.17 | 0.27 | 0.56 | - | - | - |

Relying on the learned expert weights, our method aggregates the three experts appropriately and achieves better performance on the dominant test classes, thus obtaining promising performance gains on various test distributions, as shown in Table 10. Note that the performance gain compared to existing methods gets larger as the test dataset gets more imbalanced. For example, on CIFAR100-LT with the imbalance ratio of 50, our test-time self-supervised strategy obtains a 7.7% performance gain on the Forward-LT-50 distribution and obtains a 9.2% performance gain on the Backward-LT-50 distribution, both of which are non-trivial. Such an observation is also supported by the visualization result of Figure 2, which plots the results of existing methods on ImageNet-LT with different test class distributions regarding the three class subsets.

In addition, since the imbalance degrees of the test datasets are relatively low on iNaturalist 2018, the simulated test class distributions are thus relatively balanced. As a result, the obtained performance improvement is not that significant, compared to other datasets. However, if there are more iNaturalist test samples following highly imbalanced test class distributions in real applications, our method would obtain more promising results.

Table 10: The performance improvement via test-time self-supervised aggregation on various test class distributions of ImageNet-LT, CIFAR100-LT, Places-LT and iNaturalist 2018.

| | ImageNet-LT | | | | | | | | CIFAR100-LT(IR10) | | | | | | | |
| Test Dist. | Ours w/o test-time aggregation | | | | Ours w/ test-time aggregation | | | | Ours w/o test-time aggregation | | | | Ours w/ test-time aggregation | | | |
| | Many | Med. | Few | All | Many | Med. | Few | All | Many | Med. | Few | All | Many | Med. | Few | All |
|---|---|---|---|---|---|---|---|---|---|---|---|---|---|---|---|---|
| Forward-LT-50 | 65.6 | 55.7 | 44.1 | 65.5 | 70.0 | 53.2 | 33.1 | 69.4 (+3.9) | 66.3 | 58.3 | - | 66.3 | 69.0 | 50.8 | - | 71.2 (+4.9) |
| Forward-LT-25 | 65.3 | 56.9 | 43.5 | 64.4 | 69.5 | 53.2 | 32.2 | 67.4 (+3.0) | 63.1 | 60.8 | - | 64.5 | 67.6 | 52.2 | - | 69.4 (+4.9) |
| Forward-LT-10 | 66.5 | 56.8 | 44.2 | 63.6 | 69.9 | 54.3 | 34.7 | 65.4 (+1.8) | 64.1 | 58.8 | - | 64.1 | 67.2 | 54.2 | - | 67.6 (+3.5) |
| Forward-LT-5 | 65.9 | 56.5 | 43.3 | 62.0 | 68.9 | 54.8 | 35.8 | 63.0 (+1.0) | 62.7 | 57.1 | - | 62.7 | 66.9 | 54/3 | - | 66.3 (+3.6) |
| Forward-LT-2 | 66.2 | 56.5 | 42.1 | 60.0 | 68.2 | 56.0 | 40.1 | 60.6 (+0.6) | 62.8 | 56.3 | - | 61.6 | 66.1 | 56.6 | - | 64.4 (+2.8) |
| Uniform | 67.0 | 56.7 | 42.6 | 58.8 | 66.5 | 57.0 | 43.5 | 58.8 (+0.0) | 65.5 | 59.9 | - | 63.6 | 65.8 | 58.8 | - | 63.6 (+0.0) |
| Backward-LT-2 | 66.3 | 56.7 | 43.1 | 56.8 | 65.3 | 57.1 | 45.0 | 57.1 (+0.3) | 62.7 | 56.9 | - | 60.2 | 65.6 | 59.5 | - | 62.9 (+2.7) |
| Backward-LT-5 | 66.6 | 56.9 | 43.0 | 54.7 | 63.4 | 56.4 | 47.5 | 55.5 (+0.8) | 62.8 | 57.5 | - | 59.7 | 65.1 | 60.4 | - | 62.4 (+2.7) |
| Backward-LT-10 | 65.0 | 57.6 | 43.1 | 53.1 | 60.9 | 57.5 | 50.1 | 54.5 (+1.4) | 63.5 | 58.2 | - | 59.8 | 62.5 | 61.4 | - | 61.7 (+1.9) |
| Backward-LT-25 | 64.2 | 56.9 | 43.4 | 51.1 | 60.5 | 57.1 | 50.0 | 53.7 (+2.6) | 63.4 | 57.7 | - | 58.7 | 61.9 | 62.0 | - | 62.1 (+3.4) |
| Backward-LT-50 | 69.1 | 57.0 | 42.9 | 49.8 | 60.7 | 56.2 | 50.7 | 53.1 (+3.3) | 62.0 | 57.8 | - | 58.6 | 62.6 | 62.6 | - | 63.0 (+3.8) |

| | CIFAR100-LT(IR50) | | | | | | | | CIFAR100-LT(IR100) | | | | | | | |
| Test Dist. | Ours w/o test-time aggregation | | | | Ours w/ test-time aggregation | | | | Ours w/o test-time aggregation | | | | Ours w/ test-time aggregation | | | |
| | Many | Med. | Few | All | Many | Med. | Few | All | Many | Med. | Few | All | Many | Med. | Few | All |
|---|---|---|---|---|---|---|---|---|---|---|---|---|---|---|---|---|
| Forward-LT-50 | 59.7 | 53.3 | 26.9 | 59.5 | 68.0 | 44.1 | 19.4 | 67.2 (+7.7) | 60.7 | 50.3 | 32.4 | 58.4 | 69.9 | 48.8 | 14.2 | 65.9 (+7.5) |
| Forward-LT-25 | 59.1 | 51.8 | 32.6 | 58.6 | 67.3 | 46.2 | 19.5 | 64.5 (+6.9) | 60.6 | 49.6 | 29.4 | 57.0 | 68.9 | 46.5 | 15.1 | 62.5 (+5.5) |
| Forward-LT-10 | 59.7 | 47.2 | 36.1 | 56.4 | 67.2 | 45.7 | 24.7 | 61.2 (+4.8) | 60.1 | 48.6 | 28.4 | 54.4 | 68.3 | 46.9 | 16.7 | 58.3 (+3.9) |
| Forward-LT-5 | 59.7 | 46.9 | 36.9 | 54.8 | 67.0 | 45.7 | 29.9 | 58.6 (+3.4) | 60.3 | 50.3 | 29.5 | 53.1 | 68.3 | 45.3 | 19.4 | 54.8 (+1.7) |
| Forward-LT-2 | 59.2 | 48.4 | 41.9 | 53.2 | 63.8 | 48.5 | 39.3 | 55.4 (+2.2) | 60.6 | 48.8 | 31.3 | 50.1 | 68.2 | 47.6 | 22.5 | 51.1 (+1.0) |
| Uniform | 61.0 | 50.2 | 45.7 | 53.8 | 61.5 | 50.2 | 45.0 | 53.9 (+0.1) | 61.6 | 50.5 | 33.9 | 49.4 | 65.4 | 49.3 | 29.3 | 49.8 (+0.4) |
| Backward-LT-2 | 59.0 | 48.2 | 42.8 | 50.1 | 57.5 | 49.7 | 49.4 | 51.9 (+1.8) | 61.2 | 49.1 | 30.8 | 45.2 | 63.1 | 49.4 | 31.7 | 46.2 (+1.0) |
| Backward-LT-5 | 60.1 | 48.6 | 41.8 | 48.2 | 50.0 | 49.3 | 54.2 | 50.9 (+2.7) | 62.0 | 48.9 | 32.0 | 42.6 | 56.2 | 49.1 | 38.2 | 44.7 (+2.1) |
| Backward-LT-10 | 58.6 | 46.9 | 42.6 | 46.1 | 49.3 | 49.1 | 54.6 | 51.0 (+4.9) | 60.6 | 48.2 | 31.7 | 39.7 | 52.1 | 47.9 | 40.6 | 43.9 (+4.2) |
| Backward-LT-25 | 55.1 | 48.9 | 41.2 | 44.4 | 44.5 | 46.6 | 57.0 | 51.7 (+7.3) | 58.2 | 47.9 | 32.2 | 36.7 | 48.7 | 44.2 | 41.8 | 42.5 (+5.8) |
| Backward-LT-50 | 57.0 | 48.8 | 41.6 | 43.6 | 45.8 | 46.6 | 58.4 | 52.8 (+9.2) | 66.9 | 48.6 | 30.4 | 35.0 | 49.0 | 42.7 | 42.5 | 42.4 (+7.4) |

| | Places-LT | | | | | | | | iNaturalist 2018 | | | | | | | |
| Test Dist. | Ours w/o test-time aggregation | | | | Ours w/ test-time aggregation | | | | Ours w/o test-time aggregation | | | | Ours w/ test-time aggregation | | | |
| | Many | Med. | Few | All | Many | Med. | Few | All | Many | Med. | Few | All | Many | Med. | Few | All |
|---|---|---|---|---|---|---|---|---|---|---|---|---|---|---|---|---|
| Forward-LT-50 | 43.5 | 42.5 | 65.9 | 43.7 | 46.8 | 39.3 | 30.5 | 46.4 (+2.7) | - | - | - | - | - | - | - | - |
| Forward-LT-25 | 42.8 | 42.1 | 29.3 | 42.7 | 46.3 | 38.9 | 23.6 | 44.9 (+2.3) | - | - | - | - | - | - | - | - |
| Forward-LT-10 | 42.3 | 41.9 | 34.9 | 42.3 | 45.4 | 39.0 | 27.0 | 43.3 (+1.0) | - | - | - | - | - | - | - | - |
| Forward-LT-5 | 43.0 | 44.0 | 33.1 | 42.4 | 45.6 | 40.6 | 27.3 | 42.6 (+0.2) | - | - | - | - | - | - | - | - |
| Forward-LT-2 | 43.4 | 42.4 | 32.6 | 41.3 | 44.9 | 41.2 | 29.5 | 41.3 (+0.0) | 73.9 | 72.4 | 72.0 | 72.4 | 75.5 | 72.5 | 70.7 | 72.5 (+0.1) |
| Uniform | 43.1 | 42.4 | 33.2 | 40.9 | 40.4 | 43.2 | 36.8 | 40.9 (+0.0) | 74.4 | 72.5 | 73.1 | 72.9 | 74.5 | 72.5 | 73.0 | 72.9 (+0.0) |
| Backward-LT-2 | 42.8 | 41.9 | 33.2 | 39.9 | 37.1 | 42.9 | 40.0 | 40.6 (+0.7) | 76.1 | 72.8 | 72.6 | 73.1 | 74.9 | 72.6 | 73.7 | 73.5 (+0.4) |
| Backward-LT-5 | 43.1 | 42.0 | 33.6 | 39.1 | 36.4 | 42.7 | 41.1 | 41.1 (+2.0) | - | - | - | - | - | - | - | - |
| Backward-LT-10 | 43.5 | 42.9 | 33.7 | 38.9 | 35.2 | 43.2 | 41.3 | 41.4 (+2.5) | - | - | - | - | - | - | - | - |
| Backward-LT-25 | 44.6 | 42.4 | 33.6 | 37.8 | 38.0 | 43.5 | 41.1 | 42.0 (+4.2) | - | - | - | - | - | - | - | - |
| Backward-LT-50 | 42.2 | 43.4 | 33.3 | 37.2 | 37.3 | 43.5 | 40.5 | 41.6 (+4.7) | - | - | - | - | - | - | - | - |

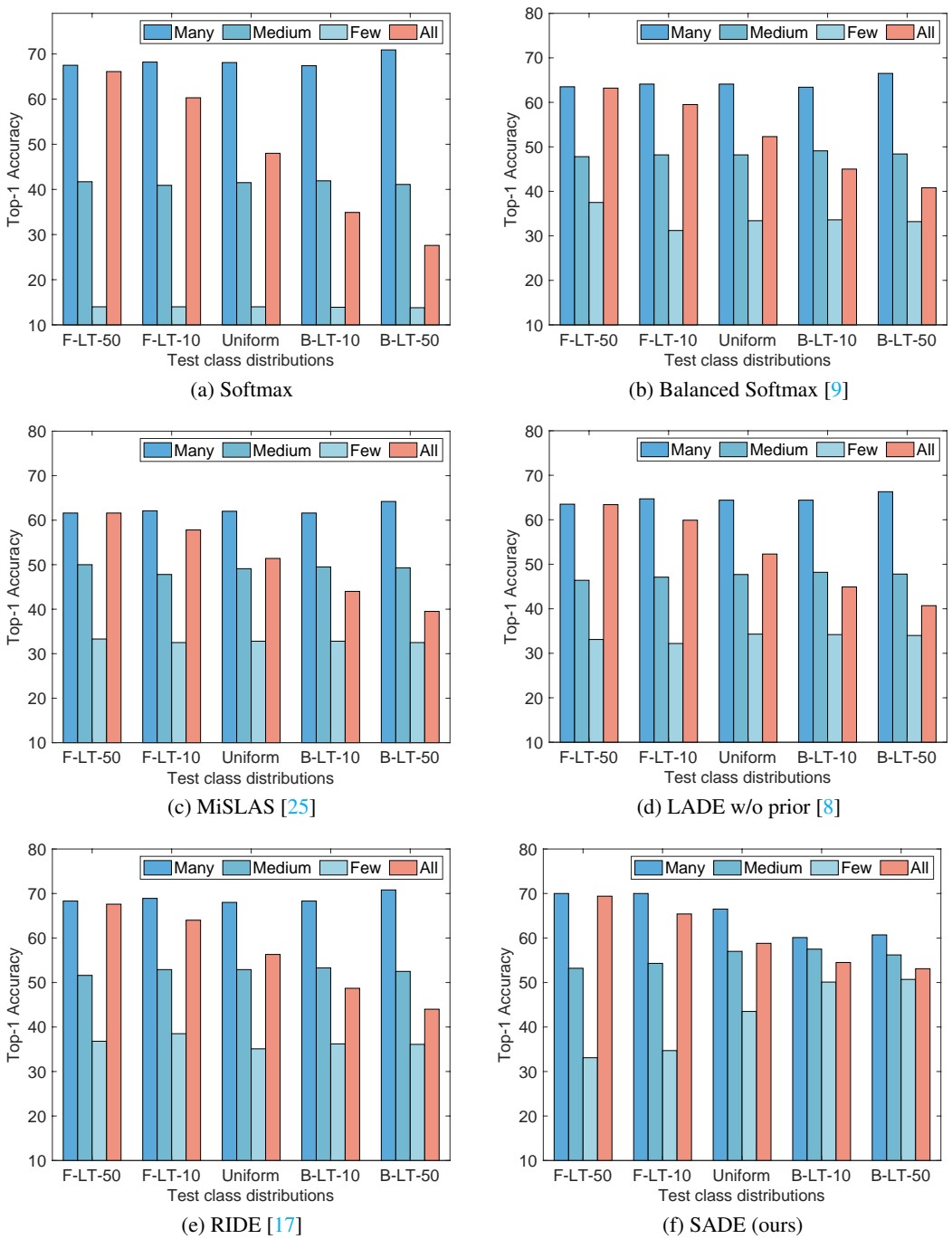

Figure 2: Top-1 accuracy of existing long-tailed (LT) methods on ImageNet-LT with various test class distributions, including uniform, forward and backward long-tailed ones with imbalance ratios 10 and 50, respectively. Here, "F-LT-$N$" and "B-LT-$N$ indicate the cases where test samples follow the same long-tailed distribution as training data and inversely long-tailed to the training data, with the imbalance ratio $N$, respectively. The results show that **existing methods perform very similarly on various test class distributions in terms of their performance on many-shot, medium-shot and few-shot classes. In contrast, our proposed method is capable of adaptingbi to various test class distributions in terms of many-shot, medium-shot and few-shot performance, thus leading to better overall performance on each test class distribution.**

# E   Ablation Studies on Skill-diverse Expert Learning

## E.1   Discussion on Expert Number

In SADE, we consider three experts, where the "forward" and "backward" experts are necessary since they span a wide spectrum of possible test class distributions, while the "uniform" expert ensures that we retain high accuracy on the uniform test class distributions. Nevertheless, our approach can be straightforwardly extended to more than three experts. For the models with more experts, we adjust the hyper-parameter $\lambda$ in Eq. (3) for the new experts and keep the hyper-parameters of the original three experts unchanged, so that different experts are skilled in different types of class distributions. Following this, we further evaluate the influence of the expert number on our method based on ImageNet. To be specific, when there are four experts, we set $\lambda = 1$ for the new expert; while when there are five experts, we set $\lambda = 0.5$ and $\lambda = 1$ for the two newly-added experts, respectively. As shown in Table 11, with the increasing number of experts, the ensemble performance of our method is improved on vanilla long-tailed recognition, *e.g.,* four experts obtain a 1.2% performance gain compared to three experts on ImageNet-LT. As a result, our method with more experts obtains consistent performance improvement in test-agnostic long-tailed recognition on various test class distributions compared to three experts, as shown in Table 12. Even so, only three experts are sufficient to handle varied test class distributions, and provide a good trade-off between performance and efficiency.

Table 11: Performance of our method with different numbers of experts on ImageNet-LT with the uniform test distribution.

| Model | 4 experts | | | | 5 experts | | | |
|---|---|---|---|---|---|---|---|---|
| | Many-shot | Medium-shot | Few-shot | All classes | Many-shot | Medium-shot | Few-shot | All classes |
| Expert $E_1$ | 69.4 | 44.5 | 16.5 | 50.3 | 69.8 | 44.9 | 17.0 | 50.7 |
| Expert $E_2$ | 66.2 | 51.5 | 32.9 | 54.6 | 68.8 | 48.4 | 23.9 | 52.9 |
| Expert $E_3$ | 55.7 | 52.7 | 46.8 | 53.4 | 66.1 | 51.4 | 22.0 | 54.5 |
| Expert $E_4$ | 44.1 | 49.7 | 55.9 | 48.4 | 56.8 | 52.7 | 47.7 | 53.6 |
| Expert $E_5$ | - | - | - | - | 43.1 | 59.0 | 54.8 | 47.5 |
| Ensemble | 66.6 | 58.4 | 46.7 | 60.0 | 68.8 | 58.5 | 43.2 | 60.4 |

Table 12: Performance of our method with different numbers of experts on various test class distributions of ImageNet-LT.

| Method | Experts | ImageNet-LT | | | | | | | | | | |
|---|---|---|---|---|---|---|---|---|---|---|---|---|
| | | Forward | | | | | Uniform | Backward | | | | |
| | | 50 | 25 | 10 | 5 | 2 | 1 | 2 | 5 | 10 | 25 | 50 |
| | 3 experts | 69.4 | 67.4 | 65.4 | 63.0 | 60.6 | 58.8 | 57.1 | 55.5 | 54.5 | 53.7 | 53.1 |
| SADE | 4 experts | 70.1 | 68.1 | 66.3 | 64.2 | 61.6 | 60.0 | 58.7 | 57.6 | 56.7 | 56.1 | 55.6 |
| | 5 experts | 70.7 | 68.9 | 66.8 | 64.5 | 62.1 | 60.4 | 58.7 | 57.2 | 56.3 | 55.6 | 54.7 |

## E.2 Hyper-parameters in Inverse Softmax Loss

This appendix evaluates the influence of the hyper-parameter $\lambda$ in the inverse softmax loss for the backward expert, where we fix all other hyper-parameters and only adjust the value of $\lambda$. As shown in Table 13, with the increase of $\lambda$, the backward expert simulates more inversely long-tailed distribution (to the training data), and thus the ensemble performance on *few-shot classes* is better. Moreover, when $\lambda \in \{2, 3\}$, our method achieves a better trade-off between head classes and tail classes, leading to relatively better overall performance on ImageNet-LT.

Table 13: Influence of the hyper-parameter $\lambda$ in the inverse softmax loss on ImageNet-LT with the uniform test distribution.

| Model | $\lambda = 0.5$ | | | |
|---|---|---|---|---|
| | Many-shot classes | Medium-shot classes | Few-shot classes | All long-tailed classes |
| Forward Expert $E_1$ | 69.1 | 43.6 | 17.2 | 49.8 |
| Uniform Expert $E_2$ | 66.4 | 50.9 | 33.4 | 54.5 |
| Backward Expert $E_3$ | 61.9 | 51.9 | 40.3 | 54.2 |
| Ensemble | 71.0 | 54.6 | 33.4 | 58.0 |

| Model | $\lambda = 1$ | | | |
|---|---|---|---|---|
| | Many-shot classes | Medium-shot classes | Few-shot classes | All long-tailed classes |
| Forward Expert $E_1$ | 69.7 | 44.0 | 16.8 | 50.2 |
| Uniform Expert $E_2$ | 65.5 | 51.1 | 32.4 | 54.4 |
| Backward Expert $E_3$ | 56.5 | 52.3 | 47.1 | 53.2 |
| Ensemble | 77.2 | 55.7 | 36.2 | 58.6 |

| Model | $\lambda = 2$ | | | |
|---|---|---|---|---|
| | Many-shot classes | Medium-shot classes | Few-shot classes | All long-tailed classes |
| Forward Expert $E_1$ | 68.8 | 43.7 | 17.2 | 49.8 |
| Uniform Expert $E_2$ | 65.5 | 50.5 | 33.3 | 53.9 |
| Backward Expert $E_3$ | 43.4 | 48.6 | 53.9 | 47.3 |
| Ensemble | 67.0 | 56.7 | 42.6 | 58.8 |

| Model | $\lambda = 3$ | | | |
|---|---|---|---|---|
| | Many-shot classes | Medium-shot classes | Few-shot classes | All long-tailed classes |
| Forward Expert $E_1$ | 69.6 | 43.8 | 17.4 | 50.2 |
| Uniform Expert $E_2$ | 66.2 | 50.7 | 33.1 | 54.2 |
| Backward Expert $E_3$ | 43.4 | 48.6 | 53.9 | 48.0 |
| Ensemble | 67.8 | 56.8 | 42.4 | 59.1 |

| Model | $\lambda = 4$ | | | |
|---|---|---|---|---|
| | Many-shot classes | Medium-shot classes | Few-shot classes | All long-tailed classes |
| Forward Expert $E_1$ | 69.1 | 44.1 | 16.3 | 49.9 |
| Uniform Expert $E_2$ | 65.7 | 50.8 | 32.6 | 54.1 |
| Backward Expert $E_3$ | 21.9 | 38.1 | 58.9 | 34.7 |
| Ensemble | 60.2 | 57.5 | 50.4 | 57.6 |

| Model | $\lambda = 5$ | | | |
|---|---|---|---|---|
| | Many-shot classes | Medium-shot classes | Few-shot classes | All long-tailed classes |
| Forward Expert $E_1$ | 69.7 | 43.7 | 16.5 | 50.0 |
| Uniform Expert $E_2$ | 65.9 | 50.9 | 33.0 | 54.2 |
| Backward Expert $E_3$ | 16.0 | 33.9 | 60.6 | 30.6 |
| Ensemble | 56.3 | 57.5 | 54.0 | 56.6 |

# F   Ablation Studies on Test-time Self-supervised Aggregation

## F.1   Influences of Training Epoch

As illustrated in Section 5.1, we set the training epoch of our test-time self-supervised aggregation strategy to 5 on all datasets. Here, we further evaluate the influence of the epoch number, where we adjust the epoch number from 1 to 100. As shown in Table 14, when the training epoch number is larger than 5, the learned expert weights by our method are converged on ImageNet-LT, which verifies that our method is robust enough. The corresponding performance on various test class distributions is reported in Table 15.

Table 14: The influence of the epoch number on the learned expert weights by test-time self-supervised aggregation on ImageNet-LT.

| Test Dist. | Epoch 1 | | | Epoch 5 | | | Epoch 10 | | |
|---|---|---|---|---|---|---|---|---|---|
| | E1 ($w_1$) | E2 ($w_2$) | E3 ($w_3$) | E1 ($w_1$) | E2 ($w_2$) | E3 ($w_3$) | E1 ($w_1$) | E2 ($w_2$) | E3 ($w_3$) |
| Forward-LT-50 | 0.44 | 0.33 | 0.23 | 0.52 | 0.35 | 0.13 | 0.52 | 0.37 | 0.11 |
| Forward-LT-25 | 0.43 | 0.34 | 0.23 | 0.50 | 0.35 | 0.15 | 0.50 | 0.37 | 0.13 |
| Forward-LT-10 | 0.43 | 0.34 | 0.23 | 0.46 | 0.36 | 0.18 | 0.46 | 0.36 | 0.18 |
| Forward-LT-5 | 0.41 | 0.34 | 0.25 | 0.43 | 0.34 | 0.23 | 0.43 | 0.35 | 0.22 |
| Forward-LT-2 | 0.37 | 0.33 | 0.30 | 0.37 | 0.35 | 0.28 | 0.38 | 0.33 | 0.29 |
| Uniform | 0.34 | 0.31 | 0.35 | 0.33 | 0.33 | 0.34 | 0.33 | 0.32 | 0.35 |
| Backward-LT-2 | 0.30 | 0.32 | 0.38 | 0.29 | 0.31 | 0.40 | 0.29 | 0.32 | 0.39 |
| Backward-LT-5 | 0.27 | 0.29 | 0.44 | 0.24 | 0.31 | 0.45 | 0.23 | 0.31 | 0.46 |
| Backward-LT-10 | 0.24 | 0.29 | 0.47 | 0.21 | 0.29 | 0.50 | 0.21 | 0.30 | 0.49 |
| Backward-LT-25 | 0.23 | 0.29 | 0.48 | 0.18 | 0.29 | 0.53 | 0.17 | 0.3 | 0.53 |
| Backward-LT-50 | 0.24 | 0.29 | 0.47 | 0.17 | 0.27 | 0.56 | 0.15 | 0.28 | 0.57 |

| Test Dist. | Epoch 20 | | | Epoch 50 | | | Epoch 100 | | |
|---|---|---|---|---|---|---|---|---|---|
| | E1 ($w_1$) | E2 ($w_2$) | E3 ($w_3$) | E1 ($w_1$) | E2 ($w_2$) | E3 ($w_3$) | E1 ($w_1$) | E2 ($w_2$) | E3 ($w_3$) |
| Forward-LT-50 | 0.53 | 0.38 | 0.09 | 0.53 | 0.38 | 0.09 | 0.53 | 0.38 | 0.09 |
| Forward-LT-25 | 0.51 | 0.37 | 0.12 | 0.52 | 0.37 | 0.11 | 0.50 | 0.38 | 0.12 |
| Forward-LT-10 | 0.44 | 0.36 | 0.20 | 0.45 | 0.37 | 0.18 | 0.46 | 0.36 | 0.18 |
| Forward-LT-5 | 0.42 | 0.35 | 0.23 | 0.42 | 0.35 | 0.23 | 0.42 | 0.35 | 0.23 |
| Forward-LT-2 | 0.38 | 0.33 | 0.29 | 0.39 | 0.33 | 0.28 | 0.38 | 0.32 | 0.30 |
| Uniform | 0.33 | 0.33 | 0.34 | 0.34 | 0.32 | 0.34 | 0.32 | 0.33 | 0.35 |
| Backward-LT-2 | 0.29 | 0.31 | 0.40 | 0.30 | 0.32 | 0.38 | 0.29 | 0.30 | 0.41 |
| Backward-LT-5 | 0.24 | 0.31 | 0.45 | 0.23 | 0.29 | 0.48 | 0.25 | 0.30 | 0.45 |
| Backward-LT-10 | 0.20 | 0.30 | 0.50 | 0.21 | 0.31 | 0.48 | 0.21 | 0.30 | 0.49 |
| Backward-LT-25 | 0.16 | 0.30 | 0.54 | 0.17 | 0.29 | 0.54 | 0.17 | 0.30 | 0.53 |
| Backward-LT-50 | 0.15 | 0.29 | 0.56 | 0.14 | 0.29 | 0.57 | 0.14 | 0.29 | 0.57 |

Table 15: The influence of the epoch number on the performance of test-time self-supervised aggregation on ImageNet-LT.

| Test Dist. | Epoch 1 | | | | Epoch 5 | | | | Epoch 10 | | | |
|---|---|---|---|---|---|---|---|---|---|---|---|---|
| | Many | Med. | Few | All | Many | Med. | Few | All | Many | Med. | Few | All |
| Forward-LT-50 | 68.8 | 54.6 | 37.5 | 68.5 | 70.1 | 53.2 | 33.1 | 69.4 | 70.1 | 52.9 | 32.4 | 69.5 |
| Forward-LT-25 | 68.6 | 54.9 | 34.9 | 66.9 | 69.5 | 53.2 | 32.2 | 67.4 | 69.7 | 52.5 | 32.5 | 67.5 |
| Forward-LT-10 | 60.3 | 55.3 | 37.6 | 65.2 | 69.9 | 54.3 | 34.7 | 65.4 | 69.9 | 54.5 | 35.0 | 65.4 |
| Forward-LT-5 | 68.4 | 55.3 | 37.3 | 63.0 | 68.9 | 54.8 | 35.8 | 63.0 | 68.8 | 54.9 | 36.0 | 63.0 |
| Forward-LT-2 | 67.9 | 56.2 | 40.8 | 60.6 | 68.2 | 56.0 | 40.1 | 60.6 | 68.2 | 56.0 | 39.7 | 60.5 |
| Uniform | 66.7 | 56.9 | 43.1 | 58.8 | 66.5 | 57.0 | 43.5 | 58.8 | 66.4 | 56.9 | 43.4 | 58.8 |
| Backward-LT-2 | 65.6 | 57.1 | 44.7 | 57.1 | 65.3 | 57.1 | 45.0 | 57.1 | 65.3 | 57.1 | 45.0 | 57.1 |
| Backward-LT-5 | 63.9 | 57.6 | 46.8 | 55.5 | 63.4 | 56.4 | 47.5 | 55.5 | 63.3 | 57.4 | 47.8 | 55.6 |
| Backward-LT-10 | 62.1 | 57.6 | 47.9 | 54.2 | 60.9 | 57.5 | 48.9 | 54.5 | 61.1 | 57.6 | 48.9 | 54.5 |
| Backward-LT-25 | 62.4 | 57.6 | 48.5 | 53.4 | 60.5 | 57.1 | 50.0 | 53.7 | 60.5 | 57.1 | 50.3 | 53.8 |
| Backward-LT-50 | 64.9 | 56.7 | 47.8 | 51.9 | 60.7 | 56.2 | 50.7 | 53.1 | 60.1 | 55.9 | 51.2 | 53.2 |

| Test Dist. | Epoch 20 | | | | Epoch 50 | | | | Epoch 100 | | | |
|---|---|---|---|---|---|---|---|---|---|---|---|---|
| | Many | Med. | Few | All | Many | Med. | Few | All | Many | Med. | Few | All |
| Forward-LT-50 | 70.3 | 52.2 | 32.4 | 69.5 | 70.3 | 52.2 | 32.4 | 69.5 | 70.0 | 52.2 | 32.4 | 69.3 |
| Forward-LT-25 | 69.8 | 52.4 | 31.4 | 67.5 | 69.9 | 52.3 | 31.4 | 67.6 | 69.7 | 52.6 | 32.6 | 67.5 |
| Forward-LT-10 | 69.6 | 54.8 | 35.8 | 65.3 | 69.8 | 54.6 | 35.2 | 65.4 | 69.8 | 54.6 | 35.0 | 65.4 |
| Forward-LT-5 | 68.7 | 55.0 | 36.4 | 63.0 | 68. | 55.0 | 36.4 | 63.0 | 68.7 | 54.7 | 36.7 | 62.9 |
| Forward-LT-2 | 68.1 | 56.0 | 39.9 | 60.5 | 68.3 | 55.9 | 39.6 | 60.5 | 68.2 | 56.0 | 40.1 | 60.6 |
| Uniform | 66.7 | 56.9 | 43.2 | 58.8 | 66.9 | 56.8 | 42.8 | 58.8 | 66.5 | 56.8 | 43.2 | 58.7 |
| Backward-LT-2 | 65.4 | 57.1 | 44.9 | 57.1 | 65.6 | 57.0 | 44.7 | 57.1 | 64.9 | 57.0 | 45.6 | 57.0 |
| Backward-LT-5 | 63.4 | 57.4 | 47.6 | 55.5 | 62.7 | 57.4 | 48.3 | 55.6 | 63.4 | 57.5 | 47.0 | 55.4 |
| Backward-LT-10 | 60.7 | 57.5 | 49.4 | 54.6 | 61.1 | 57.6 | 48.8 | 54.4 | 60.6 | 57.6 | 49.1 | 54.5 |
| Backward-LT-25 | 60.4 | 57.1 | 50.4 | 53.9 | 60.4 | 57.0 | 50.3 | 53.8 | 60.9 | 56.8 | 50.2 | 53.7 |
| Backward-LT-50 | 60.9 | 56.1 | 51.1 | 53.2 | 60.6 | 55.9 | 51.1 | 53.2 | 60.8 | 56.1 | 51.2 | 53.2 |

## F.2 Influences of Batch Size

In previous results, we set the batch size of test-time self-supervised aggregation to 128 on all datasets. In this appendix, we further evaluate the influence of the batch size on our strategy, where we adjust the batch size from 64 to 256. As shown in Table 16, with different batch sizes, the learned expert weights by our method keep nearly the same, which shows that our method is insensitive to the batch size. The corresponding performance on various test class distributions is reported in Table 17, where the performance is also nearly the same when using different batch sizes.

Table 16: The influence of the batch size on the learned expert weights by test-time self-supervised aggregation on ImageNet-LT.

| Test Dist. | Batch size 64 | | | Batch size 128 | | | Batch size 256 | | |
|---|---|---|---|---|---|---|---|---|---|
| | E1 ($w_1$) | E2 ($w_2$) | E3 ($w_3$) | E1 ($w_1$) | E2 ($w_2$) | E3 ($w_3$) | E1 ($w_1$) | E2 ($w_2$) | E3 ($w_3$) |
| Forward-LT-50 | 0.52 | 0.37 | 0.11 | 0.52 | 0.35 | 0.13 | 0.50 | 0.33 | 0.17 |
| Forward-LT-25 | 0.49 | .0.38 | 0.13 | 0.50 | 0.35 | 0.15 | 0.48 | 0.24 | 0.18 |
| Forward-LT-10 | 0.46 | 0.36 | 0.18 | 0.46 | 0.36 | 0.18 | 0.45 | 0.35 | 0.20 |
| Forward-LT-5 | 0.44 | 0.34 | 0.22 | 0.43 | 0.34 | 0.23 | 0.43 | 0.35 | 0.22 |
| Forward-LT-2 | 0.37 | 0.34 | 0.29 | 0.37 | 0.35 | 0.28 | 0.38 | 0.33 | 0.29 |
| Uniform | 0.34 | 0.32 | 0.34 | 0.33 | 0.33 | 0.34 | 0.33 | 0.32 | 0.35 |
| Backward-LT-2 | 0.28 | .032 | 0.40 | 0.29 | 0.31 | 0.40 | 0.30 | 0.31 | 0.39 |
| Backward-LT-5 | 0.24 | 0.30 | 0.46 | 0.24 | 0.31 | 0.45 | 0.25 | 0.30 | 0.45 |
| Backward-LT-10 | 0.21 | 0.30 | 0.49 | 0.21 | 0.29 | 0.50 | 0.22 | 0.29 | 0.49 |
| Backward-LT-25 | 0.17 | 0.29 | 0.54 | 0.18 | 0.29 | 0.53 | 0.20 | 0.28 | 0.52 |
| Backward-LT-50 | 0.15 | 0.30 | 0.55 | 0.17 | 0.27 | 0.56 | 0.19 | 0.27 | 0.54 |

Table 17: The influence of the batch size on the performance of test-time self-supervised aggregation on ImageNet-LT.

| Test Dist. | Batch size 64 | | | | Batch size 128 | | | | Batch size 256 | | | |
|---|---|---|---|---|---|---|---|---|---|---|---|---|
| | Many | Med. | Few | All | Many | Med. | Few | All | Many | Med. | Few | All |
| Forward-LT-50 | 70.0 | 52.6 | 33.8 | 69.3 | 70.0 | 53.2 | 33.1 | 69.4 | 69.7 | 53.8 | 34.6 | 69.2 |
| Forward-LT-25 | 69.6 | 53.0 | 33.3 | 67.5 | 69.5 | 53.2 | 32.2 | 67.4 | 69.2 | 53.7 | 32.8 | 67.2 |
| Forward-LT-10 | 69.9 | 54.3 | 34.8 | 65.4 | 69.9 | 54.3 | 34.7 | 65.4 | 69.5 | 55.0 | 35.9 | 65.3 |
| Forward-LT-5 | 69.0 | 54.6 | 35.6 | 63.0 | 68.9 | 54.8 | 35.8 | 63.0 | 68.8 | 54.9 | 36.0 | 63.0 |
| Forward-LT-2 | 68.2 | 56.0 | 40.0 | 60.6 | 68.2 | 56.0 | 40.1 | 60.6 | 68.1 | 56.0 | 40.1 | 60.5 |
| Uniform | 66.9 | 56.6 | 42.4 | 58.8 | 66.5 | 57.0 | 43.5 | 58.8 | 66.5 | 56.9 | 43.3 | 58.8 |
| Backward-LT-2 | 64.9 | 57.0 | 45.7 | 57.0 | 65.3 | 57.1 | 45.0 | 57.1 | 65.5 | 57.1 | 44.8 | 57.1 |
| Backward-LT-5 | 63.1 | 57.4 | 47.3 | 55.4 | 63.4 | 56.4 | 47.5 | 55.5 | 63.4 | 56.4 | 47.5 | 55.5 |
| Backward-LT-10 | 60.9 | 57.7 | 48.6 | 54.4 | 60.9 | 57.5 | 50.1 | 54.5 | 61.3 | 57.6 | 48.7 | 54.4 |
| Backward-LT-25 | 60.8 | 56.7 | 50.1 | 53.6 | 60.5 | 57.1 | 50.0 | 53.7 | 61.0 | 57.2 | 49.6 | 53.6 |
| Backward-LT-50 | 61.1 | 56.2 | 50.8 | 53.1 | 60.7 | 56.2 | 50.7 | 53.1 | 61.2 | 56.4 | 50.0 | 52.9 |

## F.3 Influences of Learning Rate

In this appendix, we evaluate the influence of the learning rate on our self-supervised strategy, where we adjust the learning rate from 0.001 to 0.5. As shown in Table 18, with the increase of the learning rate, the learned expert weights by our method are sharper and fit the unknown test class distributions better. For example, when the learning rate is 0.001, the weight for expert $E_1$ is 0.36 on the Forward-LT-50 test distribution, while when the learning rate increases to 0.5, the weight for expert $E_1$ becomes 0.57 on the Forward-LT-50 test distribution. Similar phenomenons are also observed on backward long-tailed test class distributions.

By observing the corresponding model performance on various test class distributions in Table 19, we find that when the learning rate is too small (*e.g.,* 0.001), our test-time self-supervised aggregation strategy is unable to converge, given a fixed training epoch number of 5. In contrast, given the same training epoch, our method can obtain better performance by reasonably increasing the learning rate.

Table 18: The influence of the learning rate on the learned expert weights by test-time self-supervised aggregation on ImageNet-LT, where the number of the training epoch is 5.

| Test Dist. | Learning rate 0.001 | | | Learning rate 0.01 | | | Learning rate 0.025 | | |
|---|---|---|---|---|---|---|---|---|---|
| | E1 ($w_1$) | E2 ($w_2$) | E3 ($w_3$) | E1 ($w_1$) | E2 ($w_2$) | E3 ($w_3$) | E1 ($w_1$) | E2 ($w_2$) | E3 ($w_3$) |
| Forward-LT-50 | 0.36 | 0.34 | 0.30 | 0.49 | 0.33 | 0.18 | 0.52 | 0.35 | 0.13 |
| Forward-LT-25 | 0.36 | 0.34 | 0.30 | 0.48 | 0.34 | 0.18 | 0.50 | 0.35 | 0.15 |
| Forward-LT-10 | 0.36 | 0.34 | 0.30 | 0.45 | 0.34 | 0.21 | 0.46 | 0.36 | 0.18 |
| Forward-LT-5 | 0.36 | 0.33 | 0.31 | 0.43 | 0.34 | 0.23 | 0.43 | 0.34 | 0.23 |
| Uniform | 0.33 | 0.33 | 0.34 | 0.34 | 0.33 | 0.33 | 0.33 | 0.33 | 0.34 |
| Backward-LT-5 | 0.31 | 0.32 | 0.37 | 0.25 | 0.31 | 0.44 | 0.24 | 0.31 | 0.45 |
| Backward-LT-10 | 0.31 | 0.32 | 0.37 | 0.22 | 0.29 | 0.49 | 0.21 | 0.29 | 0.50 |
| Backward-LT-25 | 0.31 | 0.32 | 0.37 | 0.21 | 0.28 | 0.51 | 0.18 | 0.29 | 0.53 |
| Backward-LT-50 | 0.31 | 0.32. | 0.37 | 0.20 | 0.28 | 0.52 | 0.17 | 0.27 | 0.56 |

| Test Dist. | Learning rate 0.05 | | | Learning rate 0.1 | | | Learning rate 0.5 | | |
|---|---|---|---|---|---|---|---|---|---|
| | E1 ($w_1$) | E2 ($w_2$) | E3 ($w_3$) | E1 ($w_1$) | E2 ($w_2$) | E3 ($w_3$) | E1 ($w_1$) | E2 ($w_2$) | E3 ($w_3$) |
| Forward-LT-50 | 0.53 | 0.36 | 0.11 | 0.53 | 0.37 | 0.10 | 0.57 | 0.34 | 0.09 |
| Forward-LT-25 | 0.51 | 0.36 | 0.13 | 0.52 | 0.36 | 0.12 | 0.57 | 0.34 | 0.09 |
| Forward-LT-10 | 0.45 | 0.37 | 0.18 | 0.47 | 0.36 | 0.18 | 0.44 | 0.36 | 0.20 |
| Forward-LT-5 | 0.42 | 0.35 | 0.23 | 0.47 | 0.36 | 0.18 | 0.39 | 0.36 | 0.25 |
| Uniform | 0.33 | 0.33 | 0.34 | 0.31 | 0.31 | 0.38 | 0.33 | 0.34 | 0.33 |
| Backward-LT-5 | 0.24 | 0.31 | 0.45 | 0.24 | 0.29 | 0.47 | 0.21 | 0.28 | 0.51 |
| Backward-LT-10 | 0.21 | 0.30 | 0.49 | 0.21 | 0.31 | 0.48 | 0.22 | 0.32 | 0.46 |
| Backward-LT-25 | 0.16 | 0.28 | 0.56 | 0.17 | 0.31 | 0.52 | 0.15 | 0.30 | 0.55 |
| Backward-LT-50 | 0.15 | 0.28 | 0.57 | 0.14 | 0.28 | 0.58 | 0.12 | 0.27 | 0.61 |

Table 19: The influence of learning rates on test-time self-supervised aggregation on ImageNet-LT, under training epoch 5.

| Test Dist. | Learning rate 0.001 | | | | Learning rate 0.01 | | | | Learning rate 0.025 | | | |
|---|---|---|---|---|---|---|---|---|---|---|---|---|
| | Many | Med. | Few | All | Many | Med. | Few | All | Many | Med. | Few | All |
| Forward-LT-50 | 67.3 | 56.1 | 44.1 | 67.3 | 69.5 | 54.0 | 34.6 | 69.0 | 70.0 | 53.2 | 33.1 | 69.4 |
| Forward-LT-25 | 67.4 | 56.2 | 40.3 | 66.1 | 69.2 | 53.8 | 33.2 | 67.2 | 69.5 | 53.2 | 32.2 | 67.4 |
| Forward-LT-10 | 67.7 | 56.4 | 41.9 | 64.5 | 69.6 | 55.0 | 36.1 | 65.4 | 69.9 | 54.3 | 34.7 | 65.4 |
| Forward-LT-5 | 67.2 | 55.9 | 40.8 | 62.6 | 68.7 | 55.0 | 36.2 | 63.0 | 68.9 | 54.8 | 35.8 | 63.0 |
| Uniform | 66.9 | 56.6 | 42.7 | 58.8 | 67.0 | 56.8 | 42.7 | 58.8 | 66.5 | 57.0 | 43.5 | 58.8 |
| Backward-LT-5 | 65.8 | 57.5 | 43.7 | 55.0 | 63.9 | 57.5 | 46.9 | 55.5 | 63.4 | 56.4 | 47.5 | 55.5 |
| Backward-LT-10 | 64.6 | 57.5 | 43.7 | 53.1 | 61.3 | 57.6 | 48.6 | 54.4 | 60.9 | 57.5 | 50.1 | 54.5 |
| Backward-LT-25 | 66.0 | 57.3 | 44.1 | 51.5 | 61.1 | 57.4 | 49.3 | 53.5 | 60.5 | 57.1 | 50.0 | 53.7 |
| Backward-LT-50 | 68.2 | 56.8 | 43.7 | 50.0 | 63.1 | 56.5 | 49.5 | 52.7 | 60.7 | 56.2 | 50.7 | 53.1 |

| Test Dist. | Learning rate 0.05 | | | | Learning rate 0.1 | | | | Learning rate 0.5 | | | |
|---|---|---|---|---|---|---|---|---|---|---|---|---|
| | Many | Med. | Few | All | Many | Med. | Few | All | Many | Med. | Few | All |
| Forward-LT-50 | 70.2 | 52.4 | 32.4 | 69.5 | 70.3 | 52.3 | 32.4 | 69.5 | 70.3 | 51.2 | 32.4 | 69.5 |
| Forward-LT-25 | 69.7 | 52.5 | 32.5 | 67.5 | 69.9 | 52.3 | 31.4 | 67.6 | 69.9 | 51.1 | 29.5 | 67.5 |
| Forward-LT-10 | 69.7 | 54.7 | 35.8 | 65.4 | 69.9 | 54.3 | 34.8 | 65.4 | 69.5 | 55.0 | 35.8 | 65.3 |
| Forward-LT-5 | 68.8 | 54.9 | 36.2 | 63.0 | 68.8 | 54.8 | 36.1 | 63.0 | 68.3 | 55.3 | 37.6 | 62.9 |
| Uniform | 66.6 | 56.9 | 43.2 | 58.8 | 65.6 | 57.1 | 44.7 | 58.7 | 67.8 | 56.4 | 40.9 | 58.7 |
| Backward-LT-5 | 63.6 | 57.5 | 48.9 | 55.4 | 63.0 | 57.4 | 48.1 | 55.6 | 61.4 | 57.4 | 49.2 | 55.6 |
| Backward-LT-10 | 61.1 | 57.5 | 48.9 | 54.4 | 61.3 | 57.6 | 48.6 | 54.4 | 62.0 | 57.5 | 47.9 | 54.2 |
| Backward-LT-25 | 59.9 | 56.8 | 51.0 | 53.9 | 60.9 | 57.2 | 49.9 | 53.7 | 60.2 | 56.8 | 50.8 | 53.9 |
| Backward-LT-50 | 60.1 | 56.0 | 51.2 | 53.2 | 59.6 | 55.8 | 51.3 | 53.2 | 58.2 | 55.6 | 52.2 | 53.5 |

## F.4 Results of Prediction Confidence

In our theoretical analysis (*i.e.,* Theorem 1), we find that our test-time self-supervised aggregation strategy not only simulates the test class distribution, but also makes the model predictions more confident. In this appendix, we evaluate whether our strategy can really improve the prediction confidence of models on various unknown test class distributions of ImageNet-LT. To this end, we compare the prediction confidence of our method without and with test-time self-supervised aggregation in terms of the hard mean of the highest prediction probability on all test samples.

As shown in Table 20, our test-time self-supervised aggregation strategy enables the deep model to have higher prediction confidence. For example, on the Forward-LT-50 test distribution, our strategy obtains 0.015 confidence improvement, which is non-trivial since it is an average value for a large number of samples (more than 10,000 samples). In addition, when the class imbalance ratio becomes larger, our method is able to obtain more apparent confidence improvement.

Table 20: Comparison of prediction confidence between our method without and with test-time self-supervised aggregation on ImageNet-LT, in terms of the hard mean of the highest prediction probability on each sample. The higher the highest prediction, the better the model.

| Method | Prediction confidence on ImageNet-LT | | | | | | | | | | |
| | Forward-LT | | | | | Uniform | Backward-LT | | | | |
| | 50 | 25 | 10 | 5 | 2 | 1 | 2 | 5 | 10 | 25 | 50 |
| Ours w/o test-time aggregation | 0.694 | 0.687 | 0.678 | 0.665 | 0.651 | 0.639 | 0.627 | 0.608 | 0.596 | 0.583 | 0.574 |
| Ours w test-time aggregation | 0.711 | 0.704 | 0.689 | 0.674 | 0.654 | 0.639 | 0.625 | 0.609 | 0.599 | 0.589 | 0.583 |

## F.5 Run-time Cost of Test-time Aggregation

One may be interested in the run-time cost of our test-time self-supervised aggregation strategy, so we further report its running time on Forward-LT-50 and Forward-LT-25 test class distributions for illustration. As shown in Table 21, our test-time self-supervised aggregation strategy is fast in terms of per-epoch time. The actual average additional time is only 0.009 seconds per sample at test time on V100 GPUs. The result is easy to interpret since we freeze the model parameters and only learn the aggregation weights, which is much more efficient than training the whole model. More importantly, the goal of this paper is to handle a practical yet challenging test-agnostic long-tailed recognition task. For solving this challenging problem, we believe it is acceptable to allow models to be trained more, while the promising results in previous experiments have demonstrated the effectiveness of our proposed test-time self-supervised learning strategy in handling this problem. In the future, we will further extend the proposed method for better computational efficiency, e.g., exploring dynamic network routing.

Table 21: Run-time cost of our test-time self-supervised aggregation strategy on ImageNet-LT, compared to the run-time cost of model training. Here, we show two test class distributions for illustration, which have different numbers of test samples.

| Dataset | Model training | Test-time weight learning | |
| | | Forward-LT-50 | Forward-LT-25 |
| Per-epoch time | 713 s | 110 s | 130 s |

## F.6   Test-time Self-supervised Aggregation on Streaming Test Data

In the previous experiments, we conduct the test-time strategy in an offline manner [13]. However, as mentioned in Section 4.2, our test-time strategy can also be conducted in an online manner and does not require access to all the test data in advance. To verify this, we further conduct our test-time strategy on steaming test data of ImageNet-LT. As shown in Table 22, our test-time strategy performs well on the streaming test data. Even when the test data come in one by one, our test-time self-supervised strategy still outperforms the state-of-the-art baseline (i.e., offline Tent [16]) by a large margin.

Table 22: Results of our test-time self-supervised aggregation strategy on streaming test data of ImageNet-LT, where all test-time strategies are used on the same skill-diverse multi-expert model.

| Backbone | Test-time strategy | Forward-LT | | Backward-LT | |
|---|---|---|---|---|---|
| | | 50 | 5 | 5 | 50 |
| | No test-time adaptation | 65.5 | 62.0 | 54.7 | 49.8 |
| | Offline Tent [16] | 68.0 | 62.8 | 53.2 | 45.7 |
| SADE | Offline self-supervised aggregation (ours) | 69.4 | 63.0 | 55.5 | 53.1 |
| | Online self-supervised aggregation with batch size 64 | 69.5 | 63.6 | 55.8 | 53.1 |
| | Online self-supervised aggregation with batch size 8 | 69.8 | 63.0 | 55.4 | 53.0 |
| | Online self-supervised aggregation with batch size 1 | 69.0 | 62.8 | 55.2 | 52.8 |

## G More Discussions on Model Complexity

In this appendix, we discuss the model complexity of our method in terms of the number of parameters, multiply-accumulate operations (MACs) and top-1 accuracy on test-agnostic long-tailed recognition. As shown in Table 23, both SADE and RIDE belong to ensemble-based long-tailed learning methods, so they have more parameters (about 1.5x) and MACs (about 1.4x) than the original backbone model, where we do not use the efficient expert assignment trick in [17] for both methods. Because of the ensemble effectiveness of the multi-expert scheme, both methods perform much better than non-ensemble methods (*e.g.,* Softmax and other long-tailed methods). In addition, since our method and RIDE use the same multi-expert framework, both methods have the same number of parameters and MACs. Nevertheless, by using our proposed skill-diverse expert learning and test-time self-supervised aggregation strategies, our method performs much better than RIDE with no increase in model parameters and computational costs.

One may concern the multi-expert scheme leads to more model parameters and higher computational costs than the original backbone. However, note that the main focus of this paper is to solve the challenging test-agnostic long-tailed recognition, while promising results have shown that our method addresses this problem well. In this sense, slightly increasing the model complexity is acceptable for solving this practical yet challenging problem. Moreover, since there have already been many studies [6, 19] showing effectiveness in improving the efficiency of the multi-expert scheme, we think the computation increment is not a severe issue and we leave it to the future.

Table 23: Model complexity and performance of different methods in terms of the parameter number, Multiply–Accumulate Operations (MACs) and top-1 accuracy on test-agnostic long-tailed recognition. Here, we do not use the efficient expert assignment trick in [17] for RIDE and our method.

| | | | ImageNet-LT (**ResNeXt-50**) | | | | | | | | | | |
| Method | Params (M) | MACs (G) | Forward-LT | | | | | Uniform | Backward-LT | | | | |
| | | | 50 | 25 | 10 | 5 | 2 | 1 | 2 | 5 | 10 | 25 | 50 |
| Softmax | 25.03 (1.0x) | 4.26 (1.0x) | 66.1 | 63.8 | 60.3 | 56.6 | 52.0 | 48.0 | 43.9 | 38.6 | 34.9 | 30.9 | 27.6 |
| RIDE [17] | 38.28 (1.5x) | 6.08 (1.4x) | 67.6 | 66.3 | 64.0 | 61.7 | 58.9 | 56.3 | 54.0 | 51.0 | 48.7 | 46.2 | 44.0 |
| SADE (ours) | 38.28 (1.5x) | 6.08 (1.4x) | **69.4** | **67.4** | **65.4** | **63.0** | **60.6** | **58.8** | **57.1** | **55.5** | **54.5** | **53.7** | **53.1** |
| | | | CIFAR100-LT-IR100 (**ResNet-32**) | | | | | | | | | | |
| Method | Params (M) | MACs (G) | Forward-LT | | | | | Uniform | Backward-LT | | | | |
| | | | 50 | 25 | 10 | 5 | 2 | 1 | 2 | 5 | 10 | 25 | 50 |
| Softmax | 0.46 (1.0x) | 0.07 (1.0x) | 63.3 | 62.0 | 56.2 | 52.5 | 46.4 | 41.4 | 36.5 | 30.5 | 25.8 | 21.7 | 17.5 |
| RIDE [17] | 0.77 (1.5x) | 0.10 (1.4x) | 63.0 | 59.9 | 57.0 | 53.6 | 49.4 | 48.0 | 42.5 | 38.1 | 35.4 | 31.6 | 29.2 |
| SADE (ours) | 0.77 (1.5x) | 0.10 (1.4x) | **65.9** | **62.5** | **58.3** | **54.8** | **51.1** | **49.8** | **46.2** | **44.7** | **43.9** | **42.5** | **42.4** |
| | | | Places-LT (**ResNet-152**) | | | | | | | | | | |
| Method | Params (M) | MACs (G) | Forward-LT | | | | | Uniform | Backward-LT | | | | |
| | | | 50 | 25 | 10 | 5 | 2 | 1 | 2 | 5 | 10 | 25 | 50 |
| Softmax | 60.19 (1.0x) | 11.56 (1.0x) | 45.6 | 42.7 | 40.2 | 38.0 | 34.1 | 31.4 | 28.4 | 25.4 | 23.4 | 20.8 | 19.4 |
| RIDE [17] | 88.07 (1.5x) | 13.18 (1.1x) | 43.1 | 41.8 | 41.6 | 42.0 | 41.0 | 40.3 | 39.6 | 38.7 | 38.2 | 37.0 | 36.9 |
| SADE (ours) | 88.07 (1.5x) | 13.18 (1.1x) | **46.4** | **44.9** | **43.3** | **42.6** | **41.3** | **40.9** | **40.6** | **41.1** | **41.4** | **42.0** | **41.6** |
| | | | iNaturalist 2018 (**ResNet-50**) | | | | | |
| Method | Params (M) | MACs (G) | Forward-LT | | Uniform | Backward-LT | |
| | | | 3 | 2 | 1 | 2 | 3 |
| Softmax | 25.56 (1.0x) | 4.14 (1.0x) | 65.4 | 65.5 | 64.7 | 64.0 | 63.4 |
| RIDE [17] | 39.07 (1.5x) | 5.80 (1.4x) | 71.5 | 71.9 | 71.8 | 71.9 | 71.8 |
| SADE (ours) | 39.07 (1.5x) | 5.80 (1.4x) | **72.3** | **72.5** | **72.9** | **73.5** | **73.3** |

## H Potential Limitations

One concern is that this work only focuses on long-tailed classification problems. However, we believe this is enough for a new challenging task of test-agnostic long-tailed recognition, while how to extending to object detection and instance segmentation will be explored in the future. Another potential concern is the model complexity of our method. However, as discussed in Appendix G, the computation increment is not a very severe issue, while how to further accelerate our method will be explored in future. In addition, one may also expect to evaluate the proposed method on more test class distributions. However, as shown in Section 5.3, we have demonstrated the effectiveness of our method on the uniform class distribution, the forward and backward long-tailed class distributions with various imbalance ratios, and even partial class distributions. Therefore, we believe the empirical verification is sufficient for verifying our method, and the extension to more complex test class distributions is left to the future. Furthermore, extending our proposed method to other image domains, like medical image tasks [23], will also be an interesting direction.