# OpenReview forum: "Self-Supervised Aggregation of Diverse Experts for Test-Agnostic Long-Tailed Recognition"
_NeurIPS.cc/2022/Conference — NeurIPS 2022 Accept_

### Official Review · Reviewer_rg9d · 2022-06-18

**Rating:** 5
**Confidence:** 5
**Soundness:** 2 fair
**Presentation:** 3 good
**Contribution:** 2 fair

**Summary:**

This paper studies an interesting problem in long-tailed recognition, i.e.,  the training class distribution is long-tailed while the test class distribution is agnostic rather than a uniform distribution as assumed in previous works. To deal with the problem, this paper proposes a new approach that outperforms existing methods in both vanilla and test-agnostic long-tailed recognition settings.

**Questions:**

1. Why does this paper choose Balanced softmax and a variant logit adjustment loss?
2. How should we decide the number of experts?
3. Is the classifier re-weighting strategy optimal? And can the Test-time Self-supervised Aggregation learn optimal weights?
4. Why should we use the proposed method instead of existing long-tailed methods such as RIDE? The performance improvement is not significant and the proposed method incurs additional computational costs.

**Limitations:**

Yes, the authors have adequately addressed the limitations and potential negative societal impact of their work.

**Strengths And Weaknesses:**

### Strength

1. The studies problem is interesting and under-explored.
2. Extensive experiments are conducted to justify the effectiveness of the proposed method.
3. The writing is clear and easy to understand.

## Weakness

1. The technical significance is not enough. Specifically, there are two aspects. First, **the** **skill-diverse expert learning** does not make new contribution to the field because multiple experts have been used in many existing literature, e.g., [1-3]. Moreover, the idea of aggregating multiple diverse models was also explored in [3] though the studied problem is different. Second, **the** **Test-time Self-supervised Aggregation** simply a re-weighting of three models. The key contribution might be the prediction stability maximization, but optimizing this objective does not ensure to obtain the optimal weights.
2. This paper only considers the transductive setting where the entire test data are accessible at once. However, in many applications, the assumption is not satisfied.
3. This paper incurs more computational cost than previous methods. The Test-time Self-supervised Aggregation has to be performed at each test time.
4. nitpick: some bold numbers in Table 9 are not the best results.

[1] Learning from multiple experts: Self-paced knowledge distillation for long-tailed classification

[2] Long-tailed recognition by routing diverse distribution-aware experts

[3] Cross-Domain Empirical Risk Minimization for Unbiased Long-tailed Classification

---

> ### Author Response · Authors · 2022-08-02
> **Response to Reviewer rg9d  (Part 3/3)**
>
> **Q4. Concern on incurring more computational cost by our test-time strategy**
>
> As the first feasible method to handle the challenging task of test-agnostic LT, we believe that incurring a little additional computational cost is acceptable. This is also supported  by Reviewer eBs7 ("*The problem of computation complexity seems tolerable since it is a quite challenging problem after all*"). Moreover, as clarified in the above answer to Q3, our test-time self-supervised strategy can be conducted in an online manner, which is efficient in practice. In fact, the actual average additional time is *only 0.009 seconds per sample* at test time on  V100 GPUs. In the future, we will further extend the proposed method for better computational efficiency, e.g., exploring dynamic network routing.
>
> ---
>
> **Q5. Why does this paper choose Balanced softmax and a variant logit adjustment loss?**
>
> We aim to learn multiple experts with diverse skills that excel at handling different class distributions,  while developing more sophisticated learning methods for training each expert is not our focus. Therefore,  **our first principle is to utilize the  representative loss functions which have been proved simple and effective previously**. Promising results (cf. Tables 6&18) also confirmed the effectiveness of our   strategy on the loss design in learning skill-diverse experts.
>
> ---
>
> **Q6. How should we decide the number of experts?**
>
> We set the number of experts according to the degree of whether the skill-diverse experts cover the potential Forward-LT, Uniform, and Backward-LT class distributions. Through our experiments, we found that **three experts are sufficient to handle varied unknown test class distribution**   (cf. Tables 5-6), while further adding additional experts does not lead to significant performance gains (cf. Appendix E.1).
>
>
> ---
>
> **Q7. Why should we use the proposed method instead of existing long-tailed methods such as RIDE?**
>
> Compared to existing LT methods like RIDE, our SADE method offers the following advantages:
>
> * Empirical superiority: **SADE  consistently  outperforms  existing LT methods (like RIDE) on various  test class distributions**  (cf. Table 5), including Forward LT,  Uniform, and  Backward LT  class distributions. Taking ImageNet-LT as an example, compared to RIDE, our SADE achieves *1.8%* accuracy improvement on the Forward-LT-50 test distribution, *2.5%* accuracy gain on the uniform test distribution, and *9.1%*  gain on the Backward-LT-50 test distribution.   The empirical superiority of our method has been highly recognized by both Reviewer uSnn ("*achieved consistent performance improvements on all these benchmarks  and testing scenarios*") and Reviewer eBs7 ("*The experiments and ablation studies are comprehensive and convincing*").
>
> * Theoretical aspect: **our method has a provable ability to simulate test-agnostic class distributions, while RIDE does not enjoy any theoretical guarantees**. This has also been recognized by Reviewer eBs7 ("*The proposed test-time aggregation strategy is interesting and has proven to be useful*").
>
>
> Because of the above superiority, our method provides a better solution to handle real-world long-tailed applications, where the test class distribution may follow any kind of class distribution.
>
> ---
>
> **Q8**. Typo in Table 9: Thanks for pointing this out. We have corrected the typo in the revision.
>
>
>
> Thanks again for your constructive comments. We welcome and are happy to discuss any further questions you may have.

---

> > ### Comment · Reviewer_rg9d · 2022-08-10
> > **After rebuttal**
> >
> > Thank the authors for addressing some of my concerns. To reflect the authors' efforts, I will change my score to 5.

---

> > > ### Author Response · Authors · 2022-08-10
> > > **Thank you very much for your valuable time**
> > >
> > > We sincerely appreciate your time and effort in reviewing our paper. Based on your constructive suggestions, the work has become more solid and thorough now.

---

> ### Author Response · Authors · 2022-08-02
> **Response to Reviewer rg9d (Part 2/3)**
>
> **Q2. Can the Test-time Self-supervised Aggregation learn optimal weights?**
>
> It is hard to theoretically prove that our method can find the global optimum, but it performs well in our experiments (cf. Tables 7-10). In addition, **we also  empirically find that the solution of our method is close to the optimum**.  Specifically, we conduct grid search to find   optimal weights, where the values of the three weights are selected from [0, 0.1, 0.2, 0.3, 0.4, 0.5, 0.6, 0.7, 0.8, 0.9] and the sum of them is 1. The obtained optimal weights by grid search and the corresponding model performance on ImageNet-LT are reported in the following table. We find that our self-supervised strategy is able to obtain near-optimal weights and model performance compared to the results of the grid search, which further demonstrates the effectiveness of our method. Moreover, analyzing the theoretical optimum is beyond the scope of this paper, which we thus leave as future work.
>
> | Test Distributions |          Method           | Weight of Expert 1 | Weight of Expert 2 | Weight of Expert 3 | Performance |
> | ---------------- | ----------------------- | :----------------: | :----------------: | :----------------: | :----------------: |
> |   Forward-LT-50   |        SADE (ours)        |        0.46        |        0.35        |        0.13        |    69.4     |
> |   Forward-LT-50   | **Grid search** (optimal) |        0.50        |        0.40        |        0.10        |    69.8     |
> |                    |                           |                    |                    |                    |                    |
> |   Forward-LT-10   |        SADE (ours)        |        0.46        |        0.36        |        0.18        |    65.4    |
> |   Forward-LT-10   | **Grid search** (optimal) |        0.50        |        0.40        |        0.10        |    65.7     |
> |                    |                           |                    |                    |                    |                    |
> |   Backward-LT-10   |        SADE (ours)        |        0.21        |        0.29        |        0.50        |    54.5     |
> |   Backward-LT-10   | **Grid search** (optimal) |        0.20        |        0.30        |        0.50        |    54.7     |
> |                    |                           |                    |                    |                    |                    |
> |   Backward-LT-50   |        SADE (ours)        |        0.17        |        0.27        |        0.56        |    53.1    |
> |   Backward-LT-50   | **Grid search** (optimal) |        0.10        |        0.30        |        0.60        |    53.5      |
>
> ---
>
> **Q3. Does  the proposed test-time strategy have to require all test data to be available in advance?**
>
>
> Indeed not. After submission, we also tried to apply our method for streaming test data, and found that **our test-time self-supervised strategy also works well in an online manner**, and **does not require access to all the test data in advance**. More specifically, as shown in the following table, our test-time strategy performs well on the *streaming test data* of ImageNet-LT. Even when the test data come in one by one, our test-time self-supervised strategy still outperforms the SOTA baseline (i.e., offline Tent [2])  by a large margin, with the same multi-expert model.
>
> In the revised paper, we have clarified  in Section 4.2  that our test-time strategy can be conducted in an online manner  for streaming test data, and also added the following new results  to Appendix F.6. The source code of our online strategy  will be released.
>
> |          Test-time Strategy           | Forward-LT-50 | Forward-LT-5 | Backward-LT-5 | Backward-LT-50 |
> | ----------------------------------- | :-----------: | :----------: | :-----------: | :------------: |
> |                 No test-time adaptation                   |     65.5      |     62.0     |     54.7      |      49.8      |
> |                 Entropy minimization with Tent [2]               |  68.0     |     62.8    |     53.2      |      45.7     |
> |   offline SADE (reported in paper)    |     69.4      |     63.0     |     55.5      |      53.1      |
> | online SADE with batch size 64 |     69.5      |     63.6     |     55.8      |      53.1      |
> | online SADE with batch size 8  |     69.8      |     63.0     |     55.4      |      53.0      |
> | online SADE with batch size 1  |     69.0      |     62.8     |     55.2      |      52.8      |
>
>
> [4] Tent: Fully test-time adaptation by entropy minimization. In ICLR, 2021.

---

> ### Author Response · Authors · 2022-08-02
> **Response to Reviewer rg9d  (Part 1/3)**
>
> Thank the reviewer for the constructive comments, particularly for recognizing the studied problem is interesting and under-explored. We  address the concerns point by point as follows.
>
> ---
>
> **Q1. Concern on the technical significance**
>
> We see your point, but we are afraid that we cannot agree with the comment on technical significance based on the following facts.
>
> **(Q1-1) "The technical significance is not enough."**
>
> The  test-agnostic LT problem we attempt to address  is  highly challenging  (as recognized by Reviewer eBs7). To  our best knowledge, there is no existing feasible solution  so far --- previous methods either assume the test class distribution to be fixed as uniform, or  the imbalanced  test class distribution is known *a priori*. **Our SADE is the first feasible approach to solve this problem and achieve superior performance** (cf. Table 5), which  we believe  is already a nontrivial technical contribution.
>
>
> **(Q1-2) "The skill-diverse expert learning does not make new contribution to the field because multiple experts have been used in  existing literature, e.g., [1-3]."**
>
> This is not true.  **Our proposed inverse softmax loss (cf. Eq.3)  is new   to the community**. Despite its simplicity, it effectively increases the expert diversity that  leads to higher   ensemble performance  (cf. Table 6), and enables our model to cover inverse LT class distributions, making a core component to solve the challenging  test-agnostic LT problem.
>
> **Existing multi-expert methods are not directly applicable**. Simply training and aggregating multiple experts [1,2,3] cannot  handle the challenge of unknown class distribution shifts. As shown in the following table, when the test class distribution varies, the performance distribution of the multi-expert model learned by RIDE  [2] in terms of  many-, medium- and few-shot classes  remains unchanged, suggesting that the method cannot adapt to different test class distributions.
>
>
> | Test distribution |   Method    | Many | Medium | Few   | All classes |
> | --------------------- | --------- | :--: | :----: | :--: | :--: |
> |      Forward-LT-50      |    RIDE     | 68.3 |  51.6  | 36.8 | 67.6 |
> |      Forward-LT-10      |    RIDE     | 68.9 |  52.9  | 38.5 | 64.0 |
> |         Uniform         |    RIDE     | 68.0 |  52.9  | 35.1 | 56.3 |
> |     Backward-LT-10      |    RIDE     | 68.3 |  53.3  | 36.2 | 48.7 |
> |     Backward-LT-50      |    RIDE     | 70.8 |  52.5  | 36.1 | 44.0 |
> |                         |             |      |        |      |      |
> |      Forward-LT-50      | SADE (ours) | 70.0 |  53.2  | 33.1 | 69.4 (+1.8)  |
> |      Forward-LT-10      | SADE (ours) | 69.9 |  54.3  | 34.7 | 65.4 (+1.4) |
> |         Uniform         | SADE (ours) | 66.5 |  57.0  | 43.5 | 58.8 (+2.5) |
> |     Backward-LT-50      | SADE (ours) | 60.9 |  57.5  | 50.1 | 54.5 (+5.8) |
> |     Backward-LT-50      | SADE (ours) | 60.7 |  56.2  | 50.7 | 53.1 (+9.1) |
>
>
> [1] Learning from multiple experts: Self-paced knowledge distillation for long-tailed classification. In ECCV, 2020.
>
> [2] Long-tailed recognition by routing diverse distribution-aware experts. In ICLR, 2021.
>
> [3] Cross-Domain Empirical Risk Minimization for Unbiased Long-tailed Classification. In AAAI, 2022.
>
> **(Q1-3) "The Test-time Self-supervised Aggregation is simply a re-weighting of three models. The idea of aggregating multiple diverse models was also explored in [3]."**
>
>
> Our test-time self-supervised strategy is not a simple expert aggregation strategy. To the best of our knowledge,  **maximizing prediction consistency  between    unlabeled test data's perturbed views for expert aggregation is new**. Such a strategy is  well-motivated  (cf. Table 2), theoretically guaranteed (cf. Theorem 1)  and empirically effective (cf. Tables 5&8) in handling unknown class distribution shifts.   The novelty of this strategy has been highly recognized by Reviewer eBs7 "*The proposed test-time aggregation strategy is interesting and has proven to be useful*".
>
> In addition, please note that existing LT methods for  aggregating multiple experts are unable to tackle test-agnostic LT. For example, the mentioned method [3] uses **ground-truth labels to compute the weights for different experts**. However, when facing test-agnostic LT, this method does not make sense anymore, because the ground-truth labels are unavailable at test time.
>
>
> In light of the above   technical innovations we have introduced, it is not fair to criticize the technical significance of our method just because of its simplicity.

---

### Official Review · Reviewer_eBs7 · 2022-07-09

**Rating:** 7
**Confidence:** 5
**Soundness:** 3 good
**Presentation:** 3 good
**Contribution:** 3 good

**Summary:**

This paper extends the conventional long-tailed learning to the "test-agnostic one", in which the model trained on a long-tailed class distribution should generalize to arbitrary testing distribution not necessarily being uniform. To handle such a problem, this paper proposes a novel method consisting of two modules: (1) diverse experts with different class expertise and (2) a self-supervised test-time weighting strategy that adaptively aggregates the experts to tackle unknown testing distribution. Extensive experiments have shown the efficacy of the proposed method on handling arbitrary class distributions in testing.


**Questions:**

Line 762: how exactly is the expertise-guided loss functions changed to suit different types of distributions?


**Limitations:**

The limitations are carefully discussed in the paper, which mainly encompass the extensibility to different tasks and the model complexity of the proposed method.


**Strengths And Weaknesses:**

## Strengths
- The proposed test-agnostic setting is challenging and of great practical significance.
- The paper is well-written and easy to follow.
- The proposed test-time aggregation strategy is interesting and has proven to be useful.
- The experiments and ablation studies are comprehensive and convincing.

## Weaknesses
- The weakness is mainly focused on the computation complexity. As mentioned in the paper, the three experts are independent in ResNet blocks (later stages) and fully-connected layers. Though it seems tolerable since it is a quite challenging problem after all, have the authors explored the trade-off between accuracy and complexity? For instance, is it a near-linear relationship between higher accuracy and experts with fewer shared modules? I would like to see how far can it go at the two extreme points: (1) when nothing is shared between experts and (2) everything is shared except the fully-connected layers.

---

> ### Author Response · Authors · 2022-08-02
> **Response to Reviewer eBs7**
>
> Thank you very much for your encouraging comments on our paper, particularly for recognizing the value of the studied problem and our proposed method. We hope that our work can motivate more future long-tailed studies to tackle this practical yet challenging problem, i.e., unknown test class distribution shifts. We   address your questions as follows.
>
> ---
>
> **Q1. Trade-off between computation complexity and performance.  Whether is it a near-linear relationship between higher accuracy and experts with fewer shared modules?**
>
> We are glad to see the reviewer agrees that the additional computation complexity is acceptable since our studied problem is challenging. As mentioned in Appendix C.3 (Lines 634-640), we also made efforts to reduce the computational costs by sharing the majority of the model backbone between  experts. We only set the top network blocks of ResNet/ResNeXt  and the classifier as independent components of each expert, and reduce  their number of convolutional channels by 1/4. We found this design provides a good computation-performance trade-off.
>
> Here, we further evaluate the relationship between the number of shared model blocks and model performance based on ImageNet-LT under the same hyper-parameter setting. As shown in the following table, **the relationship between the number of the shared blocks and model accuracy is not near linear**. **Sharing two blocks is already a good trade-off** between model accuracy and total computational complexity (in terms of MACs) on ImageNet-LT.
>
>
> |         Model         | MACs (G) | Forward-LT-50 | Forward-LT-25 | Uniform | Backward-LT-25 | Backward-LT-50 |
> | ------------------- | :-------------: | :-----------: | :-----------: | :-----: | :------------: | :------------: |
> |   Share all blocks    |      3.29       |     65.9      |     64.0      |  52.9   |      49.7      |      49.8      |
> |    Share first 3 blocks     |      4.27       |     69.0      |     67.0      |  58.0   |      53.1      |      52.5      |
> | Share first 2 blocks (ours) |      6.08       |     69.4      |     67.4      |  58.8   |      53.7      |      53.1      |
> |    Share first 1 block     |      8.33       |     69.2      |     67.7      |  59.0   |      53.9      |      53.4      |
> |     Share 0 block     |      9.64       |     68.9      |     66.9      |  58.9   |      53.4      |      52.9      |
>
>
>
>
> ---
>
> **Q2. How exactly are the expertise-guided loss functions changed for more experts in Appendix E.1 (Line 762)?**
>
> In the experiments of having more experts in Appendix E.1, **we adjusted the hyper-parameter $\lambda$ in Eq. (3) for  new experts, while keeping the hyper-parameters of the original three experts unchanged**. Specifically, when there are four experts, we set $\lambda=1$ for the new expert; while when there are five experts, we set $\lambda=0.5$ and $\lambda=1$ for the two newly-added experts, respectively. In the revised paper, we have clarified this in  Appendix E.1.
>
>
>
> We are glad to discuss any further questions you may have.

---

### Official Review · Reviewer_uSnn · 2022-07-09

**Rating:** 5
**Confidence:** 5
**Soundness:** 2 fair
**Presentation:** 3 good
**Contribution:** 2 fair

**Summary:**

The aim of this paper is to develop a mixture-of-expert (MOE) model to solve the test-agnostic long-tailed recognition problem, where the test class distribution may follow a uniform, forward or backward long-tailed distribution. The method is developed on the basis of RIDE with three experts and consists of two strategies. At training time, SADE utilizes skill-diverse expert learning strategies that require each expert to handle a different class distribution in order to solve distribution-agnostic long-tailed recognition problems. At test time, SADE utilizes a test-time expert aggregation strategy, which is based on a self-supervised learning approach, to determine expert aggregation methods that handle unknown class distributions. Experiments were conducted on various test-time training strategies dealing with class distribution transfer. SADE achieves state-of-the-art performance on multiple long-tailed datasets, including CIFAR100-LT, ImageNet-LT, Places-LT and iNaturalist 2018.

**Questions:**

I have a few questions on the fairness of the result comparisons and the setting of the test-time adaptation:
- When comparing with the current SOTA method RIDE [46], the results of RIDE (https://github.com/frank-xwang/RIDE-LongTailRecognition/blob/main/MODEL_ZOO.md) is actually achieved by training the model for 100 epochs. However, the results reported in the paper is achieved by training the model for 200 epochs. Therefore, I am concerned about the fairness of the comparisons.
- For test-time adaptation, have you tried using MC-Dropout [1] as a metric for expert aggregation? You can get the uncertainty of each expert and decide the weight of each expert based on the uncertainty. Does it save you from using all test data for expert aggregation? I think it might help if MC-Dropout is used with a strong data augmentation to produce different inputs.

[1] Gal, Yarin, and Zoubin Ghahramani. "Dropout as a bayesian approximation: Representing model uncertainty in deep learning." international conference on machine learning. PMLR, 2016.

**Ethics Review Area:**

["I don’t know"]

**Limitations:**

Yes, the authors have adequately addressed the limitations and potential negative societal impact of their work. The long-tail recognition algorithm aims to alleviate the problem of ignoring underrepresented minorities, which is necessary to obtain an unbiased model and facilitate the development of CNN models for social justice.

**Strengths And Weaknesses:**

Strengths:
- Evaluating on Forward-LT and Uniform test-class distributions can help us better understanding the performance of various long-tailed algorithms on different testing scenarios. It is shown that SADE can achieve SOTA performance on all these testing distributions.
- This paper is well-written and easy to follow.
- Authors conducted thorough experiments on various benchmarks and testing scenarios and achieved consistent performance improvements on all these benchmarks.

For weaknesses, my main concerns are twofold:
- The biggest improvement comes from test data with Backward-LT distributions, however, I find it hard to believe that backward class distributions are common in real-world applications. The many-shot (few-shot) classes in the training data are often the many-shot (few-shot) classes in the testing data as well. Thus, Forward-LT and Uniform test class distributions make more sense, however, SADE achieves marginal improvements in these two testing cases.
- The test-time self-supervised aggregation strategy requires the model to see all test data (unlabeled) before deployment, however, in real-world applications we more commonly see only 1 test image. This part is similar to my first concern, which is whether this setup is a practical setup that can be used in real-world applications.

---

> ### Author Response · Authors · 2022-08-02
> **Response to Reviewer uSnn (Part 2/2)**
>
> **Q3. Does the proposed test-time strategy have to use all test data for test-time training?**
>
> Indeed not. After submission, we also tried to apply our method for streaming test data, and found that **our test-time self-supervised strategy also works well in an online manner**, and **does not require access to all the test data in advance**. More specifically, as shown in the following table, our test-time strategy performs well on the *streaming test data* of ImageNet-LT. Even when the test data come in one by one, our test-time self-supervised strategy still outperforms the SOTA baseline (i.e., offline Tent [2])  by a large margin, with the same multi-expert model.
>
> In the revised paper, we have clarified  in Section 4.2  that our test-time strategy can be conducted in an online manner  for streaming test data, and also added the following new results  to Appendix F.6. The source code of our online strategy  will be released.
>
> |          Test-time Strategy           | Forward-LT-50 | Forward-LT-5 | Backward-LT-5 | Backward-LT-50 |
> | ----------------------------------- | :-----------: | :----------: | :-----------: | :------------: |
> |                 No test-time adaptation                   |     65.5      |     62.0     |     54.7      |      49.8      |
> |                 Entropy minimization with Tent [2]               |     68.0     |     62.8    |     53.2      |      45.7      |
> |   offline SADE (reported in paper)    |     69.4      |     63.0     |     55.5      |      53.1      |
> |  online  SADE with batch size 64 |     69.5      |     63.6     |     55.8      |      53.1      |
> |  online  SADE with batch size 8  |     69.8      |     63.0     |     55.4      |      53.0      |
> |  online  SADE with batch size 1  |     69.0      |     62.8     |     55.2      |      52.8      |
>
> [2] Tent: Fully test-time adaptation by entropy minimization. ICLR, 2021.
>
> ---
>
> **Q4. Implementation of RIDE and the fairness of comparisons**
>
> We used the same setup for all the baselines and our method, where **RIDE was also trained for 200 epochs on ImageNet-LT** based on its official code (https://github.com/frank-xwang/RIDE-LongTailRecognition). Thus the empirical comparisons between RIDE and our method are fair. The following table shows the reproduced results of RIDE based on 100 and 200 training epochs on ImageNet-LT.
>
> |      Methods      | Many-shot | Medium-shot | Few-shot | All classes  |
> | :---------------: | :-------: | :---------: | :------: | :--: |
> | RIDE - 100 epochs |   67.4    |    52.5     |   34.5   | 55.8 |
> | RIDE - 200 epochs |   68.0    |    52.9     |   35.1   | 56.3 |
>
> ---
>
> **Q5. Exploration on MC-Dropout**
>
> Thanks for your constructive suggestion. We did not consider MC-Dropout for expert aggregation before. Following your suggestion, we further explore MC-Dropout to estimate the uncertainty of the trained experts and use the uncertainty as a metric to aggregate experts at test time. The results based on three experts on ImageNet-LT are shown in the following table. We find that **using MC-Dropout as an aggregation metric provides reasonable performance, but still worse than our self-supervised aggregation strategy, particularly when the test imbalance ratio is large**. Such a result further demonstrates the superiority of our method. We   agree that exploring model uncertainty to aggregate experts is interesting, and will explore it in the future.
>
>
>
> | Test-time Strategy | Forward-LT-50 | Forward-LT-25 | Uniform  | Backward-LT-25 | Backward-LT-50 |
> | :----------------: | :-----------: | :-----------: | :------: | :------------: | :------------: |
> |         No         |     65.5      |     64.4      | **58.8** |      51.5      |      49.8      |
> |     MC-Dropout     |     67.8      |     66.4      |   58.7   |      52.2      |      51.0      |
> |    SADE (ours)     |   **69.4**    |   **67.4**    | **58.8** |    **53.7**    |    **53.1**    |
>
>
>
>
> We welcome and are happy to discuss any further questions.

---

> > ### Comment · Reviewer_uSnn · 2022-08-09
> > **Keep my rating as "Borderline accept"**
> >
> > I would like to thank the authors for addressing my concerns and conducting additional experiments during the short rebuttal period. I will keep my rating as "Borderline accept".

---

> > > ### Author Response · Authors · 2022-08-09
> > > **Thanks for the response**
> > >
> > > Thanks very much for the response. We have tried our best to address all the mentioned concerns. Following your constructive suggestions, our work has become more solid and thorough now. Could you please kindly re-evaluate the work based on the current version? We would like to know whether there is any remaining question that we can resolve.

---

> ### Author Response · Authors · 2022-08-02
> **Response to Reviewer uSnn (Part 1/2)**
>
> Thanks a lot for your valuable comments. We are glad to see that the thorough experiments and good writing of this paper are appreciated. We address your concerns point by point as follows.
>
> ---
>
> **Q1. Concern on the rationality of test-agnostic LT**
>
> **Please note that test-agnostic LT is not an entirely new task**. As stated in Lines 37-38, it has been explored by LADE [1] (published in CVPR 2021) where the test class distribution can be Forward LT, Uniform, or  Backward LT distributions. However, LADE assumes that the test class distribution is known *a priori*, which does not hold in realistic applications. This makes  LADE less applicable in practice.
>
> **Our main contribution is to eliminate such a restrictive assumption made by LADE and explore test-agnostic LT under the setting of unknown test class distributions**. Hence, the developed solutions would have more practical value, and may motivate future LT research. The value of our task has been highly recognized by both Reviewer eBs7 "*The proposed test-agnostic setting is challenging and of great practical significance*" and Reviewer rg9d "*The studies problem is interesting and under-explored*".
>
> In addition, an example of backward class distributions in realistic scenarios is autonomous driving, which has been described in Lines 32-36 of our submission.
>
>
> [1] Disentangling label distribution for long-tailed visual recognition. In CVPR, 2021.
>
> ---
>
> **Q2. The performance improvement on Forward-LT and Uniform is not significant**
>
> Thanks for recognizing "*the proposed method achieved consistent performance improvements on all benchmarks and testing scenarios*". Please note that **improving the model performance for ALL kinds of test class distributions is nontrivial, since tackling unknown class distribution shifts between training and test data  is highly challenging**. To the best of our knowledge, no existing long-tailed method can address this problem, and our work provides the first feasible solution (cf. Tables 5&8-10).
>
> Besides, the forward-LT and uniform settings have been extensively studied. Even compared with the strong SOTA methods like RIDE and LADE, our SADE method can still bring at least *2.5%* (for uniform) and  *1.8%* (for forward-LT-50) accuracy improvement, which we believe is significant w.r.t. such strong baselines.
>
> We would like to highlight that our method is advantageous in handling large class distribution shifts between the training and test data, as mentioned in Lines 305-306 ("*the performance advantages of SADE become larger   as the test data  get  more imbalanced*"). For the forward-LT setting, when the test imbalance ratio becomes more severe, SADE demonstrates  larger performance gains, as shown in the following table.
>
>
> |   Method    | Forward-LT-500 | Forward-LT-300 | Forward-LT-200 | Forward-LT-100 | Forward-LT-50 |
> | --------- | :------------: | :------------: | :------------: | :------------: | :-----------: |
> |    RIDE     |      70.2      |      69.8      |      69.0      |      68.6      |     67.6      |
> | SADE (ours) |  74.3 (+4.1)   |  73.9 (+4.1)   |  72.7 (+3.7)   |  72.2 (+3.6)   |  69.4 (+1.8)  |

---

### Author Response · Authors · 2022-08-06
**General Response**

We sincerely appreciate all reviewers‘ time and effort in reviewing our paper and providing constructive feedback. Besides the response to each reviewer,  here we would like to further 1) thank reviewers for their recognition of our work, 2) highlight the new results added during the rebuttal, and 3) highlight the revision in the revised paper:

**1) We are glad that the reviewers appreciate and recognize our contributions.**

* The proposed test-agnostic long-tailed recognition setting is challenging, interesting, and of great practical significance [eBs7,rg9d]
* The proposed test-time aggregation strategy is interesting and has proven to be useful [eBs7]
* The experiments and ablation studies are comprehensive, convincing, and thorough. [uSnn,eBs7,rg9d]
* This paper is well-written and easy to follow. [uSnn,eBs7,rg9d]


**2) In the rebuttal, we have provided more supporting results following the reviewers’ suggestions.**

* Performance of our test-time self-supervised strategy on streaming test data [uSnn,rg9d]
* Performance of our test-time self-supervised strategy on Forward-LT test  class distributions with higher  imbalance ratios   [uSnn]
* Performance of using MC-Dropout as a metric for test-time expert aggregation [uSnn]
* Comparison results between model accuracy and the number of shared blocks between experts   [eBs7]
* Comparison with the optimal expert weights searched by Grid Search [rg9d]
* Actual average additional time of test-time aggregation per sample  [rg9d]

**3) We make the following modifications in our revision to address reviewers' questions (highlighted in blue).**

* We further clarify that our self-supervised aggregation strategy can be conducted in an online manner for streaming test data, and add corresponding empirical verification  [uSnn,rg9d]
* We add more implementation details [uSnn,eBs7]
* We add more explanations for Appendix E.1  [eBs7]



Since the discussion period will end in a few days, we would like to know whether there is any remaining question that we can resolve. We look forward to your response.

---

### Meta-Review · Area_Chair_yJVG · 2022-08-22

**Recommendation:** Accept
**Confidence:** Certain

**Metareview:**

The reviewers agreed the paper provides some nice insights into tackling the difficult and under-explored problem of test-agnostic long-tailed recognition. The reviewers appreciated the thorough experiments and ablations provided. The author response sufficiently addressed the key concerns the reviewers had.

**Award:**

No

---

### Decision · Program_Chairs · 2022-09-14

Accept